# Differentiation signals from glia are fine-tuned to set neuronal numbers during development

Anadika R Prasad, Inês Lago-Baldaia, Matthew P Bostock, Zaynab Housseini, Vilaiwan M Fernandes*

Department of Cell and Developmental Biology, University College London, London, United Kingdom

**Abstract** Neural circuit formation and function require that diverse neurons are specified in appropriate numbers. Known strategies for controlling neuronal numbers involve regulating either cell proliferation or survival. We used the *Drosophila* visual system to probe how neuronal numbers are set. Photoreceptors from the eye-disc induce their target field, the lamina, such that for every unit eye there is a corresponding lamina unit (column). Although each column initially contains ~6 post-mitotic lamina precursors, only 5 differentiate into neurons, called L1-L5; the 'extra' precursor, which is invariantly positioned above the L5 neuron in each column, undergoes apoptosis. Here, we showed that a glial population called the outer chiasm giant glia (xg$^O$), which resides below the lamina, secretes multiple ligands to induce L5 differentiation in response to epidermal growth factor (EGF) from photoreceptors. By forcing neuronal differentiation in the lamina, we uncovered that though fated to die, the 'extra' precursor is specified as an L5. Therefore, two precursors are specified as L5s but only one differentiates during normal development. We found that the row of precursors nearest to xg$^O$ differentiate into L5s and, in turn, antagonise differentiation signalling to prevent the 'extra' precursors from differentiating, resulting in their death. Thus, an intricate interplay of glial signals and feedback from differentiating neurons defines an invariant and stereotyped pattern of neuronal differentiation and programmed cell death to ensure that lamina columns each contain exactly one L5 neuron.

**\*For correspondence:**
vilaiwan.fernandes@ucl.ac.uk

## Editor's evaluation

This manuscript describes how control over the induction of neuronal fate from a progenitor pool regulates the generation of the appropriate numbers of neurons in the developing *Drosophila* retina. It describes how this occurs non-autonomously through two distinct glial cell types. It will be of interest to cell and developmental biologists and neuroscientists.

## Introduction

Many sensory systems consist of repeated circuit units that map stimuli from the outside world onto sequential processing layers (*Luo and Flanagan, 2007*). It is critical that both absolute and relative neuronal numbers are carefully controlled for these circuits to assemble with topographic correspondence across processing layers. Neuronal numbers can be set by controlling how many progeny a neural stem cell produces, or by regulating how many neural progeny survive (*Hidalgo and ffrench-Constant, 2003*; *Miguel-Aliaga and Thor, 2009*). To investigate other developmental strategies that set neuronal numbers, we used the highly ordered and repetitive *Drosophila melanogaster* visual system. Like vertebrate visual systems, the fly visual system is organised retinotopically into repeated

modular circuits that process sensory input from unique points in space spanning the entire visual field (*Hadjieconomou et al., 2011*; *Malin and Desplan, 2021*).

Retinotopy between the compound eye and the first neuropil in the optic lobe, the lamina, is built during development. Photoreceptors are born progressively in the eye imaginal disc as a wave of differentiation sweeps across the tissue from posterior to anterior. Newly born photoreceptors express Hedgehog (Hh), which promotes further wave propagation (*Treisman, 2013*). They also express the epidermal growth factor (EGF), Spitz (Spi), which recruits additional photoreceptors into developing ommatidia (*Treisman, 2013*). As photoreceptors are born, their axons project into the optic lobe and induce the lamina, such that there is a corresponding lamina unit (or cartridge) for every ommatidium (*Figure 1A*; *Hadjieconomou et al., 2011*). Each cartridge is composed of five interneurons (L1-L5; named for the medulla layers they project to) and multiple glial subtypes (*Fischbach and Dittrich, 1989*; *Hadjieconomou et al., 2011*).

Lamina induction is a multi-step process triggered by photoreceptor-derived signals. Photoreceptor-derived Hh converts neuroepithelial cells into lamina precursor cells (LPCs), promotes their terminal divisions and drives the assembly of lamina pre-cartridges referred to as columns, that is, ensembles of ~6 post-mitotic LPCs stacked together and associated with photoreceptor axon bundles (*Figure 1A and B*; *Huang and Kunes, 1998*; *Huang and Kunes, 1996*; *Sugie et al., 2010*; *Umetsu et al., 2006*). Once assembled into columns, LPCs are diversified by graded Hh signalling along the distal-proximal axis of young columns (*Bostock et al., 2022*). They then differentiate into neurons following an invariant spatio-temporal pattern whereby the most proximal (bottom) and most distal (top) cells differentiate first into L5 and L2, respectively; differentiation then proceeds in a distal-to-proximal (top-to-bottom) sequence, L3 forming next, followed by L1, then L4 (*Fernandes et al., 2017*; *Huang et al., 1998*; *Tan et al., 2015*). The sixth LPC, located between L4 and L5, does not differentiate but instead is fated to die by apoptosis and is later cleared (*Figure 1A*; *Apitz and Salecker, 2014*). This spatio-temporal pattern of neuronal differentiation is driven in part by a population of glia called wrapping glia, which ensheathes photoreceptor axons and which induces L1-L4 neuronal differentiation via insulin/insulin-like growth factor signalling in response to EGF from photoreceptors (*Fernandes et al., 2017*). Intriguingly, L1-L4 neuronal differentiation can be disrupted by manipulating wrapping glia without affecting L5 differentiation (*Fernandes et al., 2017*). Indeed, the mechanisms that drive L5 differentiation are not known. Importantly, we do not understand how exactly five neuron types differentiate from six LPCs; in other words, how are lamina neuronal numbers set?

Here, we sought to determine the mechanisms that drive L5 differentiation as well as those that set neuronal numbers in the lamina. We found that a population of glia located proximal to the lamina, called the outer chiasm giant glia (xg$^O$), induces L5 neuronal differentiation in response to EGF from photoreceptors. We showed that the xg$^O$ secrete multiple signals, including the EGF Spi and a type IV Collagen, Collagen type IV alpha 1 (Col4a1), which activate mitogen-activated protein kinase (MAPK) signalling in the most proximal row of LPCs (i.e., the row of LPCs nearest to xg$^O$), thus driving their differentiation into L5s and promoting their survival. Further, we found that the 'extra' LPCs normally fated to die are specified with L5, but not L1-L4, identity. Since the most proximal row of LPCs are in closest proximity to the xg$^O$, they receive differentiation cues from xg$^O$ first and differentiate into L5s. In turn, these newly induced L5s secrete high levels of Argos (Aos), an antagonist of Spi (*Freeman et al., 1992*), to limit MAPK activity in the 'extra' LPCs thus preventing their differentiation, and leading to their death and clearance. Thus, we highlight a new mode by which neuronal numbers can be set – not only by regulating the number of neurons born or the number that survive, but also by regulating the number induced to differentiate from a larger pool of precursors. Altogether, our results indicate that the sterotyped pattern of neuronal differentiation and programmed cell death in the lamina are determined by the architecture of the developing tissue together with feedback from newly differentiating neurons.

## Results

### L5 neuronal differentiation requires EGF receptor activity in xg$^O$

We showed previously that wrapping glia induce L1-L4 neuronal differentiation in response to EGF from photoreceptors, but that L5 differentiation was regulated independently by an unknown mechanism (*Fernandes et al., 2017*). We speculated that another glial population may be involved in

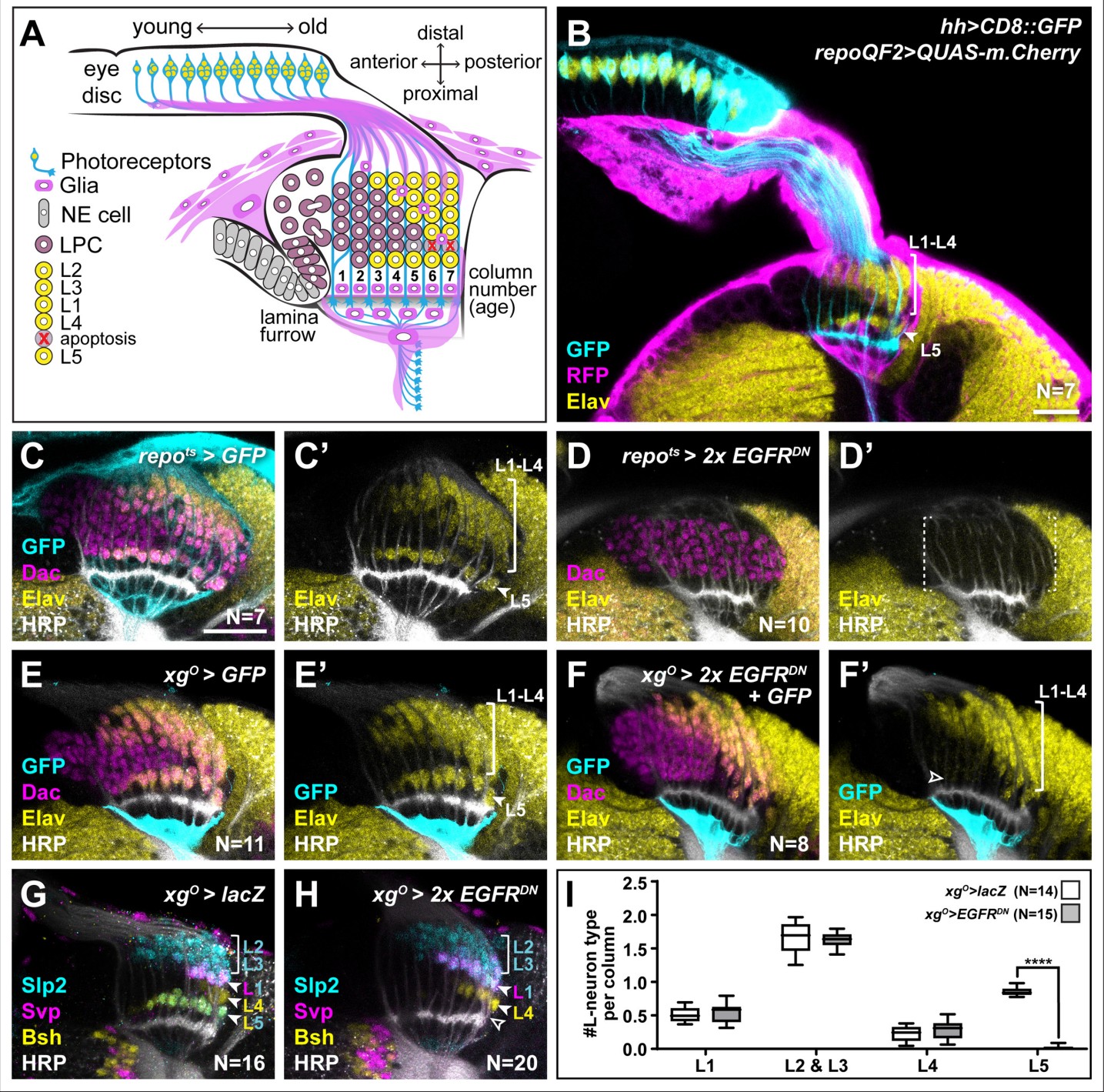

**Figure 1.** Epidermal growth factor receptor (EGFR) activity in the xg[O] is required for the differentiation of L5 neurons. (**A**) Schematic of the developing lamina. Photoreceptors (blue) drive lamina precursor cell (LPC; purple) birth from neuroepithelial cells (NEs; grey) and their assembly into columns of ~6 LPCs, which differentiate into the L1-L5 neurons (yellow) following an invariant spatio-temporal pattern. The 'extra' LPC is cleared by apoptosis (red X). Several glial types (magenta) associate with the lamina. (**B**) A cross-sectional view of an early pupal (0–5 hr after puparium formation; APF) optic lobe where *hh-Gal4* drives *UAS-CD8::GFP* expression in photoreceptors (cyan). The pan-glial driver *repo-QF2* drives *QUAS-m.Cherry* (magenta) in all glia. Embryonic lethal abnormal vision (Elav) (yellow) marks all neurons. (**C**) A cross-sectional view of an optic lobe with pan-glial expression of CD8::GFP stained for GFP (cyan), Dachshund (Dac) (magenta), Elav (yellow), and Horseradish Peroxidase (HRP; axons; white). (**D**) Pan-glial expression of two copies of EGFR[DN] stained for Dac (magenta), Elav (yellow), and HRP (white). (**E**) xg[O]-specific expression of CD8::GFP stained for GFP (cyan), Dac (magenta), Elav (yellow), and HRP (white). (**F**) xg[O]-specific expression of two copies of EGFR[DN] and CD8::GFP stained for GFP (cyan), Dac (magenta), Elav (yellow), and HRP (white). The number of Elav+ cells in proximal row (L5s) decreased (empty arrowhead) relative to control (**E**). (**G,H**) HRP (white) and L-neuron-

*Figure 1 continued on next page*

*Figure 1 continued*

type-specific markers Sloppy paired 2 (Slp2) (cyan), Brain-specific homeobox (Bsh) (yellow), and Seven-up (Svp) (magenta) in (**G**) control *xg^O^>lacZ* optic lobe and (**H**) *xg^O^>2xEGFR^DN^*. L2s and L3s express Slp2; L1s express Slp2 and Svp; L4s express Bsh and L5s express Bsh and Slp2. (**I**) Quantification of the number of L-neuron types per column for control and *xg^O^>2xEGFR^DN^*. Only L5 neurons were decreased significantly ($p^{L5}$<0.0001; Mann-Whitney U-test. Ns indicated in parentheses. Boxes indicate the lower and upper quartiles; the whiskers represent the minimum and maximum values; the line inside the box indicates the median). Scale bar = 20 µm.

The online version of this article includes the following figure supplement(s) for figure 1:

**Figure supplement 1.** A Gal4 screen identifies xg^O^ as the glial subtype that regulates L5 neuronal differentiation.

inducing L5 differentiation in response to EGF from photoreceptors. To test this hypothesis, we blocked EGF receptor (EGFR) signalling in all glia using a pan-glial driver to express a dominant negative form of EGFR (*Repo>EGFR^DN^*). Although LPCs (Dac+ cells) still formed and assembled into columns, there was a complete block in lamina neuron differentiation as seen by the absence of the pan-neuronal marker, Embryonic lethal abnormal vision (Elav); that is, L5 differentiation was disrupted in addition to the differentiation of L1-L4 as expected (*Figure 1C and D*). Thus, EGFR activity in a glial population other than the wrapping glia is required for L5 neuronal differentiation.

Many glial types infiltrate the lamina (*Figure 1—figure supplement 1A*; *Chotard and Salecker, 2007*; *Edwards et al., 2012*). Therefore, we performed a screen using glia subtype-specific Gal4s to block EGFR signalling and determined what effect this manipulation had on L5s using Elav expression in the proximal lamina (*Figure 1—figure supplement 1B-M*; summarised in *Supplementary file 1*). Blocking EGFR signalling in the xg^O^ led to a dramatic reduction in the number of L5s (*Figure 1E and F*). To rule out early developmental defects, we used a temperature-sensitive Gal80 (Gal80^ts^) and shifted animals from the permissive temperature to the restrictive temperature to limit EGFR^DN^ expression in xg^O^ to begin from the third larval instar, when lamina development initiates. This resulted in a similar loss of Elav positive cells in the proximal lamina as when EGFR^DN^ was expressed continuously in the xg^O^, indicating that this phenotype is not due to an early defect in xg^O^ (*Figure 1—figure supplement 1N*). Xg^O^ are located below the lamina plexus, often with just one or two glial cells spanning the entire width of the lamina. While xg^O^ extend fine processes towards the lamina, they do not appear to contact LPCs or L5 neurons (*Figure 1—figure supplement 1O*). Importantly, blocking EGFR signalling in the xg^O^ did not affect xg^O^ numbers or morphology (*Figure 1E and F*, *Figure 1—figure supplement 1O-R*).

Since our screen used Elav expression in the proximal lamina to assess for the presence of L5s, we next examined lamina neuron-type markers to assess whether blocking EGFR activity in xg^O^ affected L5 neurons specifically. We used antibodies against Sloppy paired 2 (Slp2), Brain-specific homeobox (Bsh), and Seven-up (Svp) in combination to distinguish lamina neuron types: L2s and L3s express Slp2 alone, L1s co-express Svp and Slp2, L4s express Bsh alone, and L5s co-express Bsh and Slp2 (*Figure 1G*; *Fernandes et al., 2017*; *Hasegawa et al., 2013*; *Tan et al., 2015*). We found that the number of L5 neurons decreased specifically, while the number of all the other neuron types were unaffected (*Figure 1G–I*; $p^{L5}$ <0.0001, Mann-Whitney U-test). Finally, to test whether the absence of L5s simply reflected a developmental delay in differentiation, we examined adult optic lobes using a different L5 neuronal marker, POU domain motif 3 (Pdm3) (*Tan et al., 2015*). Similar to our results in the developing lamina, L5s were mostly absent in the adult lamina when EGFR was blocked in xg^O^ compared with controls (*Figure 1—figure supplement 1S, T*; $N^{exp}$ = 10; $N^{ctrl}$ = 11), indicating that the loss of L5s observed during development is not due to delayed induction. Thus, EGFR activity in xg^O^ is required for L5 neuronal differentiation.

## LPCs that fail to differentiate as L5s are eliminated by apoptosis

The loss of L5 neurons when EGFR was blocked in xg^O^ could be explained either by a defect in neuronal differentiation or by an earlier defect in LPC formation or recruitment to columns. To distinguish between these possibilities, we counted the number of LPCs per column when EGFR signalling was blocked in xg^O^ compared to controls (*Figure 2A–C*). For these and later analyses we considered the youngest column located adjacent to the lamina furrow to be the first column, with column number (and age) increasing towards the posterior side (*Figure 1A*). In columns 1–4, there were no differences in the number of LPCs when EGFR was blocked in xg^O^, indicating that LPC formation and column assembly occurred normally (*Figure 2C*), supporting the hypothesis that in response to EGFR

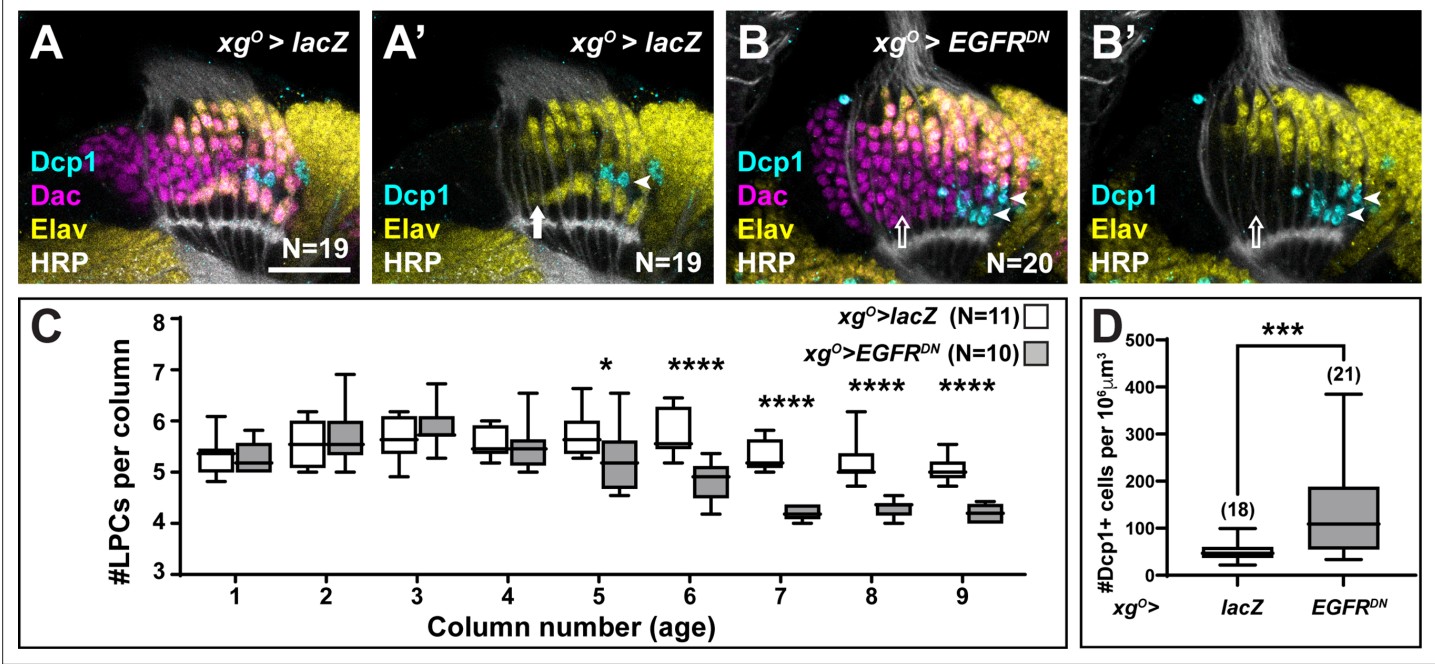

**Figure 2.** Lamina precursor cells (LPCs) that fail to differentiate into L5s undergo apoptosis. (**A**) Control *xg^O>lacZ* optic lobe stained for Death caspase-1 (Dcp-1) (cyan), Embryonic lethal abnormal vision (Elav) (yellow), and Horseradish Peroxidase (HRP) (white). Dcp-1+ cells were always observed just distal to the most proximal row of cells (L5s). (**B**) *xg^O>EGFR^DN* stained for Dcp-1 (cyan), Dachshund (Dac) (magenta), Elav (yellow), and HRP (white). Dcp-1 positive cells were observed in the most proximal row of LPCs as well as the row just distal to these. (**C**) Quantification of the number of LPCs/column (i.e., Dac+ cells/column) for control and *xg^O>EGFR^DN*. *p<0.05, ****p<0.0002; Mann-Whitney U-test. Ns indicated in parentheses. (**D**) Quantification of the number of Dcp-1 positive cells in (**A**) compared to (**B**). ***p<0.0005, Mann-Whitney U-test. Ns indicated in parentheses. Boxes indicate the lower and upper quartiles; the whiskers represent the minimum and maximum values; the line inside the box indicates the median. Scale bar = 20 μm.

activity, $xg^O$ induce proximal LPCs to differentiate as L5s. Interestingly, the number of LPCs began to decrease in older columns (column 5 onwards) when EGFR signalling was blocked in $xg^O$ (**Figure 2C**; *p<0.05, ***p<0.0002, Mann-Whitney U-test). This observation suggested that undifferentiated LPCs in older columns were being eliminated. We wondered whether LPCs that failed to differentiate into L5s underwent apoptosis, similar to the 'extra' LPCs that undergo apoptosis in controls. We used an antibody against the cleaved form of Death caspase-1 (Dcp-1), an effector caspase, to detect apoptotic cells (**Akagawa et al., 2015**) and, indeed, observed a significant increase in the number of Dcp-1 positive cells in the lamina when EGFR signalling was blocked in the $xg^O$ (132.8 cells/unit volume±19.48 standard error of the mean) compared to controls (49.14 cells/unit volume±4.53) (**Figure 2A–B and D**, p<0.0005, Mann-Whitney U-test). Importantly, we observed Dcp-1 positive cells in the proximal row of the lamina (**Figure 2B**; N^exp = 20/20), which we never observed in controls (**Figure 2A**, N^ctrl = 19/19). Altogether these results showed that EGFR activity in $xg^O$ induces the differentiation of L5 neurons, and proximal LPCs that fail to receive appropriate cues from $xg^O$ die by apoptosis.

## $xg^O$ respond to EGF from photoreceptors and secrete multiple ligands to induce MAPK-dependent neuronal differentiation of L5s

Since EGF from photoreceptors triggers EGFR activity in wrapping glia (**Fernandes et al., 2017**), we tested whether photoreceptor-derived EGF contributed to activating EGFR in $xg^O$ also. Spi is initially produced as an inactive transmembrane precursor (mSpi) that needs to be cleaved into its active secreted form (sSpi) (**Tsruya et al., 2002**). This requires the intracellular trafficking protein Star and Rhomboid proteases (**Tsruya et al., 2002**; **Urban et al., 2002**; **Yogev et al., 2008**). We took advantage of a mutant for *rhomboid 3 (rho3)* in which photoreceptors are specified but cannot secrete EGF from their axons (**Yogev et al., 2010**), resulting in failure of L1-L4 neurons to differentiate along with a significant decrease in the number of L5s (**Figure 3A and C**; p^rho3 <0.0001; one-way ANOVA with Dunn's multiple comparisons test) (**Fernandes et al., 2017**; **Yogev et al., 2010**). This result suggested that EGFR signalling in the $xg^O$ could be activated by EGF secreted by photoreceptor axons. To test

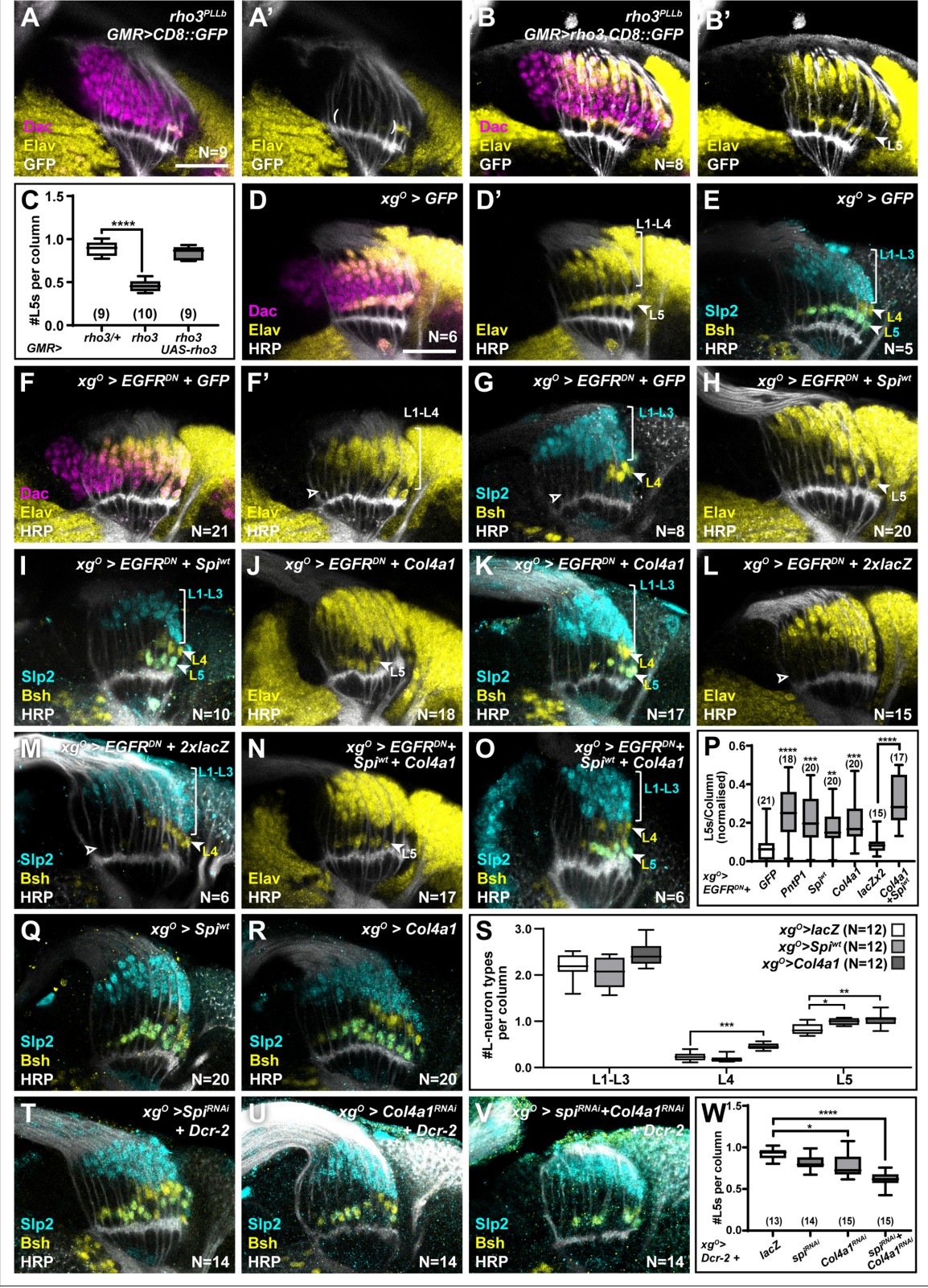

**Figure 3.** Xg$^O$ secrete multiple ligands to induce L5 neuronal differentiation in response to epidermal growth factor (EGF) from photoreceptors. (**A**) GMR-Gal4-driven CD8::GFP expression in photoreceptors in a *rho3$^{PLLb}$* background stained for GFP (white), Dachshund (Dac) (magenta), Embryonic lethal abnormal vision (Elav) (yellow). Few proximal Elav+ cells (L5s) were recovered in older columns only as previously published (***Fernandes et al., 2017***). (**B**) GMR-Gal4-driven Rho3 and CD8::GFP in a *rho3$^{PLLb}$* background stained for GFP (white), Dac (magenta), Elav (yellow) showed that L5

*Figure 3 continued on next page*

*Figure 3 continued*

neuronal differentiation was rescued (Elav+ cells in the proximal lamina). (**C**) Quantifications for number of L5 neurons/column in (**A**) and (**B**) compared to *rho3^PLLb* heterozygotes (*rho3/+*). ****p<0.0001, one-way ANOVA with Dunn's multiple comparisons test. Ns indicated in parentheses. (**D,E**) Control *xg^O>GFP* optic lobes stained for (**D**) Dac (magenta), Elav (yellow), and Horseradish Peroxidase (HRP) (white) or (**E**) HRP (white) and L-neuron-specific markers Sloppy paired 2 (Slp2) (cyan) and Brain-specific homeobox (Bsh) (yellow). (**F,G**) Gal4 titration control *xg^O>GFP + EGFR^DN* stained for (**F**) Dac (magenta), Elav (yellow), and HRP (white) or (**G**) HRP (white) and L-neuron-specific markers Slp2 (cyan) and Bsh (yellow). (**H,I**) Wild-type Spitz (Spi) (Spi^wt) co-expression with EGFR^DN specifically in xg^O stained for (**H**) Elav (yellow) and HRP (white) or (**I**) HRP (white) and L-neuron-specific markers Slp2 (cyan) and Bsh (yellow). (**J,K**) Col4a1 co-expression with EGFR^DN specifically in xg^O stained for (**J**) Elav (yellow) and HRP (white) or (**K**) HRP (white) and L-neuron-specific markers Slp2 (cyan) and Bsh (yellow). (**L,M**) Gal4 titration control *xg^O>EGFR^DN + 2xlacZ* stained for (**L**) Elav (yellow) and HRP (white) or (**M**) HRP (white), Slp2 (cyan), and Bsh (yellow). (**N,O**) Wild-type Spi^wt and Col4a1 co-expression with EGFR^DN specifically in xg^O. (**N**) stained for Elav (yellow) and HRP (white) or (**O**) HRP (white) and L-neuron-specific markers Slp2 (cyan) and Bsh (yellow). (**P**) Quantification of the number of L5s/column for the genotypes indicated compared to the appropriate titration control. For *pntP1*, *spi^wt*, and *Col4a1* co-expression with EGFR^DN, the titration control is *xg^O>EGFR^DN + GFP* (**p<0.005, ***p<0.0005; ****p<0.0001; one-way ANOVA with Dunn's multiple comparisons test. Ns indicated in parentheses). For *spi^wt* and *Col4a1* simultaneous co-expression with EGFR^DN, the titration control is *xg^O>EGFR^DN + 2xLacZ* (****p<0.0001, Mann-Whitney U-test. Ns indicated in parentheses). (**Q,R**) Optic lobes stained for Slp2 and Bsh when xg^O overexpress (**Q**) *spi^wt* or (**R**) *Col4a1.* (**S**) Quantification of the number of L-neuron types/column in (**Q**) and (**R**) compared to controls, *xg^O>lacZ*. (*p<0.05; **p<0.005; ***p<0.001; one-way ANOVA with multiple comparisons test). (**T, U, V**) Optic lobes stained for Slp2, Bsh, and HRP when xg^O co-express Dcr-2 with (**T**) spi^RNAi, (**U**) Col4a1^RNAi, and (**V**) Spi^RNAi and Col4a1^RNAi simultaneously. (**W**) Quantifications of the number of L5s/column for genotypes indicated compared to the titration control *xg^O>Dcr-2+lacZ* (*p<0.05, ****p<0.0001, one-way ANOVA with Dunn's multiple comparisons test. Scale bar = 20 μm. For all quantifications boxes indicate the lower and upper quartiles; the whiskers represent the minimum and maximum values; the line inside the box indicates the median).

The online version of this article includes the following source data and figure supplement(s) for figure 3:

**Source data 1.** Excel file containing all the probe sequences used for *in situ* hybridisation chain reaction in this study.

**Figure supplement 1.** Multiple xg^O secreted ligands activate mitogen-activated protein kinase (MAPK) signalling to drive L5 neuronal differentiation.

**Figure supplement 2.** Spi and Col4a1 from xg^O promote cell survival in proximal lamina precursor cells (LPCs).

---

this hypothesis, we restored expression of wild-type Rho3 only in photoreceptors in *rho3* mutant animals using a photoreceptor-specific driver (*GMR-Gal4*). Rho3 function in photoreceptors was sufficient to fully rescue not only L1-L4 neuronal differentiation, as previously reported (*Yogev et al., 2010*), but also L5 neuronal differentiation (*Figure 3B and C*; one-way ANOVA with Dunn's multiple comparisons test). Since photoreceptor-derived EGF was insufficient to induce L5 neuronal differentiation when EGFR signalling was blocked in xg^O (*Figure 1F and H*), together these results suggest that xg^O likely respond to EGF from photoreceptors and relay these signals to induce differentiation of proximal LPCs into L5 neurons.

We next asked what signal(s) the xg^O secrete to induce L5 differentiation. Previously, we showed that MAPK signalling is necessary and sufficient for neuronal differentiation in the lamina (*Fernandes et al., 2017*). Therefore, we reasoned that xg^O-derived differentiation signal(s) must activate MAPK signalling through a receptor tyrosine kinase (RTK) in the proximal lamina. Indeed, blocking EGFR signalling in xg^O led to reduced levels of double phosphorylated MAPK (dpMAPK) specifically in the proximal lamina (*Figure 3—figure supplement 1A, B*). The *Drosophila* genome encodes 22 ligands which activate 10 RTKs upstream of MAPK signalling (*Sopko and Perrimon, 2013*). To identify the signal(s) secreted by xg^O, we misexpressed candidate ligands and screened for their ability to rescue the loss of L5s caused by blocking EGFR activity in the xg^O. To validate this approach, we tested whether autonomously restoring transcriptional activity downstream of MAPK in xg^O while blocking EGFR activity could rescue L5 differentiation. While blocking EGFR in xg^O resulted in laminas containing 0.063±0.014 L5s per column, co-expressing PntP1 with EGFR^DN in xg^O rescued the number of L5s per column to 0.213±0.025 (*Figure 3F*, *Figure 3—figure supplement 1C* ****p<0.0001 compared to EGFR^DN alone). We then screened 18 RTK ligands based on available reagents (*Figure 3—figure supplement 1C*). Four ligands, Spi, Branchless (Bnl), Thisbe (Ths), and Col4a1, produced statistically significant rescues when compared with the *xg^O>EGFR^DN + CD8::GFP* (Gal4 titration control) (*Figure 3—figure supplement 1C*; *p<0.05, **p<0.005, ***p<0.0005, ****p<0.0001 one-way ANOVA with Dunn's multiple comparisons test). To eliminate false positive hits, we determined whether these ligands were expressed in xg^O under physiological conditions. Using a previously validated *bnl-Gal4* (*Chen and Krasnow, 2014*; *Kamimura et al., 2006*; *Spéder and Brand, 2014*; *Tamamouna et al., 2021*), we drove CD8::GFP expression and found that it was expressed in all cells of the optic lobe (*Figure 3—figure supplement 1D*), making it unlikely to be a viable hit. We found that a previously validated *ths-Gal4* (*Anllo and DiNardo, 2022*; *Wu et al., 2017*) drove CD8::GFP expression

in photoreceptors but not xg$^O$ (*Figure 3—figure supplement 1E*) consistent with previous reports (*Franzdóttir et al., 2009*). However, when we examined *Col4a1* expression using a previously validated Gal4 enhancer trap (*Hennig et al., 2006*), we found that it drove CD8::GFP expression in xg$^O$ (*Figure 3—figure supplement 1F*). We also found that a *spi-Gal4* (NP0289-Gal4; not previously validated) drove CD8::GFP expression in xg$^O$, but not photoreceptors or other cell types where *spi* is also known to be expressed, suggesting that this Gal4 line may report *spi* expression partially (*Figure 3—figure supplement 1G*). To further substantiate these results we performed fluorescence *in situ* hybridisation chain reaction (HCR), a form of fluorescent *in situ* hybridisation (*Choi et al., 2018*; *Choi et al., 2016*; *Duckhorn et al., 2022*), and confirmed that *spi* and *Col4a1* mRNAs were present in the xg$^O$ under physiological conditions (*Figure 3—figure supplement 1H and I*; see Materials and methods). This enabled us to narrow down our hits to two ligands: the EGF Spi and Col4a1, a type IV Collagen, which both rescued L5 differentiation resulting in laminas with 0.147±0.024 and 0.17±0.0197 L5s per column, respectively (*Figure 3F–K and P* p$^{spi-wt}$ <0.01 and p$^{Col4a1}$ <0.0005, one-way ANOVA with Dunn's multiple comparisons test; *Figure 3—figure supplement 1C*). Note that expressing either sSpi or wild-type (unprocessed) mSpi (referred to as Spi$^{wt}$) in xg$^O$ rescued L5 numbers (*Figure 3—figure supplement 1C*), indicating that xg$^O$ are capable of processing mSpi into the active form (sSpi).

We ruled out the trivial explanation that the rescue of L5 numbers by Spi was caused by autocrine EGFR reactivation in the xg$^O$, as Spi expression in xg$^O$ did not autonomously rescue dpMAPK nuclear localisation in xg$^O$ when EGFR signalling was blocked (*Figure 3—figure supplement 1A, B, J, K*). We then tested whether xg$^O$ express *spi* and *Col4a1* downstream of EGFR activity. We measured *spi* and *Col4a1* transcript levels using *in situ* HCR in controls and when we blocked EGFR signalling in xg$^O$. Disrupting EGFR signalling in xg$^O$ resulted in a significantly reduced fluorescence signal for *spi* and *Col4a1* transcripts in xg$^O$ compared with controls (*Figure 3—figure supplement 1H and I*, 1L-O; p$^{spi}$ <0.01, p$^{Col4a1}$<0.005; Mann-Whitney U-test). Thus, xg$^O$ express *spi* and *Col4a1* in response to EGFR activity.

Col4a1 is thought to activate MAPK signalling through its putative receptor, the Discoidin domain receptor (Ddr) (*Sopko and Perrimon, 2013*). We used a Gal4 enhancer trap in the *Ddr* locus (not previously validated) to drive CD8::GFP expression and observed that GFP was expressed in all LPCs (*Figure 3—figure supplement 1P*). We confirmed these results using *in situ* HCR, which also detected *Ddr* expression throughout the lamina (*Figure 3—figure supplement 1Q*). Spi activates EGFR (*Sopko and Perrimon, 2013*), which was shown to be expressed in LPCs previously (*Huang et al., 1998*). Thus, LPCs express the RTKs that make them competent to respond to the EGF Spi and Col4a1 produced by xg$^O$. Moreover, expressing *spi* or *Col4a1* in xg$^O$ in which EGFR signalling was blocked rescued dpMAPK signal in L5s, indicating that, when expressed in xg$^O$, these ligands were sufficient to activate MAPK signalling in the proximal lamina (*Figure 3—figure supplement 1R-T*; **p<0.005, ****p<0.0001; one-way ANOVA with Dunn's multiple comparisons test). Co-expressing Spi and Col4a1 in the *xg$^O$>EGFR$^{DN}$* background led to an enhanced and statistically significant rescue relative to individual ligand rescues alone, resulting in laminas with 0.267±0.025 L5s per column (*Figure 3L–P*; p<0.0001, Mann-Whitney U-test). We also tested whether these ligands could induce ectopic L5 differentiation when overexpressed in the xg$^O$. Overexpressing either Spi or Col4a1 resulted in a 19%±1.8 (p<0.05) and a 24%±4 (p<0.005) increase in the number of L5s per column relative to controls, respectively (*Figure 3Q–S*). Thus, Spi and Col4a1 from xg$^O$ are sufficient to induce L5 differentiation.

Next, to test whether xg$^O$-derived Spi and Col4a1 are normally required to induce L5 neuronal differentiation, we disrupted their expression specifically in xg$^O$. We used RNA interference (RNAi) to knock down *spi* and *Col4a1* expression both individually and simultaneously in xg$^O$ using previously validated lines (*Chen et al., 2016*; *Csordás et al., 2020*; *Louradour et al., 2017*; *Morante et al., 2013*; *Pastor-Pareja and Xu, 2011*). While knocking down *spi* led to a mild decrease in L5 numbers, which was not statistically significant, knocking down *Col4a1* in the xg$^O$ led to a statistically significant decrease in L5s (0.78±0.03 L5s per column) relative to controls (0.92±0.02 L5s per column) (*Figure 3T, U and W*; *p<0.05 one-way ANOVA with Dunn's multiple comparisons test). However, knocking down both *spi* and *Col4a1* simultaneously in xg$^O$ led to a strong decrease in L5s (0.61±0.02 L5s per column; *Figure 3V–W*; ****p<0.0001, one-way ANOVA with Dunn's multiple comparisons test). Under these conditions we also observed Dcp-1 positive apoptotic cells in the most proximal row of the lamina, which were never observed in controls (*Figure 3—figure supplement 2*) but were observed when L5 differentiation was blocked above (*Figure 2B*). Thus, xg$^O$-derived Spi and Col4a1 are both necessary

and sufficient to induce L5 differentiation. Altogether, we found that xg$^O$ secrete multiple factors that lead to activation of the MAPK cascade in the proximal lamina to induce differentiation of L5s.

## The 'extra' LPCs are specified as L5s though fated to die

We recently showed that a gradient of Hh signalling activity in lamina columns specifies L1-L5 identities such that high levels specify L2 and L3 (distal cell) identities, intermediate levels specify L1 and L4 (intermediate cell) identities, and low levels specify L5 (proximal cell) identity (*Bostock et al., 2022*). Since overexpressing *spi* and *Col4a1* in the xg$^O$ resulted in ectopic L5 neurons, we wondered what the source of these ectopic cells was. We quantified other lamina neuron types when either *spi* or *Col4a1* was overexpressed in xg$^O$ and found no decrease in the number of L1-L3s (Slp2-only expressing cells) or L4s (Bsh-only expressing cells) per column compared to controls (*Figure 3S*). Thus, ectopic L5s were not produced at the expense of other lamina neuron types. In wild-type optic lobes, each lamina column contains an 'extra' LPC, which is located immediately distal to the LPC fated to differentiate as an L5. These 'extra' LPCs do not differentiate but instead undergo apoptosis and are eliminated (*Figures 2A and 4A*). We hypothesised that though fated to die, 'extra' LPCs are specified with L5 identity through low Hh signalling activity in the proximal lamina, and that the overexpression of Spi and Col4a1 in xg$^O$ generated ectopic L5s by inducing differentiation and survival of the 'extra' LPCs. To test this hypothesis, we forced neuronal differentiation throughout the lamina by expressing an activated form of MAPK (MAPK$^{ACT}$) (*Figure 4—figure supplement 1A-D*) or by overexpressing the MAPK transcriptional effector, Pointed P1 (PntP1), in the lamina (*Figure 4A–D*) . As reported previously, hyperactivating MAPK signalling in the lamina led to premature neuronal differentiation: instead of sequential differentiation of L1-L4, seen as a triangular front, most lamina columns differentiated simultaneously ( (*Figure 4*) , *Figure 4—figure supplement 1A, C*; *Fernandes et al., 2017*). We observed no LPCs that remained undifferentiated (Dac+ and Elav-) past lamina column 5, including the row of cells that normally correspond to the 'extra' LPCs (*Figure 4C*, *Figure 4—figure supplement 1A, C*). Importantly, we also observed a concomitant decrease in cleaved Dcp-1 positive cells (*Figure 4C and E* ; p<0.0001, Mann-Whitney U-test), suggesting that forcing the 'extra' LPCs to differentiate blocked their death.

Next, we examined the distribution of lamina neuron types when we forced neuronal differentiation. We often observed two rows of cells co-expressing Slp2 and Bsh in the proximal lamina (*Figure 4B and D*, *Figure 4—figure supplement 1B, D*), indicating the presence of ectopic L5s. To distinguish between premature and ectopic differentiation, we quantified the number of lamina neuron types (L1-L3, L4, and L5) per column in older columns (column 7 onwards, once mature columns were observed in controls, *Figure 4—figure supplement 1E*). While there was no significant difference between the average number of L1-L3s or L4s per column, the average number of L5s per column was ~1.4-fold higher in laminas in which differentiation was ectopically induced compared with controls, that is, they contained 1.4±0.08 L5s per column compared to 1.00±0.05 L5s per column in controls (*Figure 4B, D and F–G*; p<0.0001, Mann-Whitney U-test). Thus, hyperactivating MAPK signalling in the lamina drove ectopic differentiation of L5 neurons. Importantly, ectopic L5s were only observed in the proximal but never in the distal lamina (*Figure 4D*, N=18/18; *Figure 4—figure supplement 1D*, N=9/9). Taken together, the absence of cell death in the row distal to L5s and the presence of ectopic L5s in this row indicate that hyperactivating MAPK signalling induces the 'extra' LPCs to differentiate into L5s. Thus, the 'extra' LPCs are specified as L5s though fated to die normally. These data are consistent with our work showing that lamina precursors are specified by Hh signalling prior to differentiation and that the most proximal cells, which experience the lowest levels of Hh pathway activity and are specified as L5s (*Bostock et al., 2022*). Importantly, the presence of ectopic L5s when differentiation is induced demonstrates that more LPCs are specified as L5s than differentiate normally.

## Newly born L5 neurons inhibit differentiation of distal neighbours to set neuronal number

If the two most proximal cells in each lamina column are both specified as L5s, how then is L5 differentiation limited to only the most proximal row in response to diffusible signals secreted by xg$^O$? We tested whether the 'extra' LPCs differentiated as L5s when apoptosis was blocked in animals mutant for *Death regulator Nedd2-like caspase (Dronc)*, an initiator caspase essential for caspase-dependent cell death (*Fuchs and Steller, 2011*). Cleaved Dcp-1 was absent in homozygous *Dronc$^{I24}$*

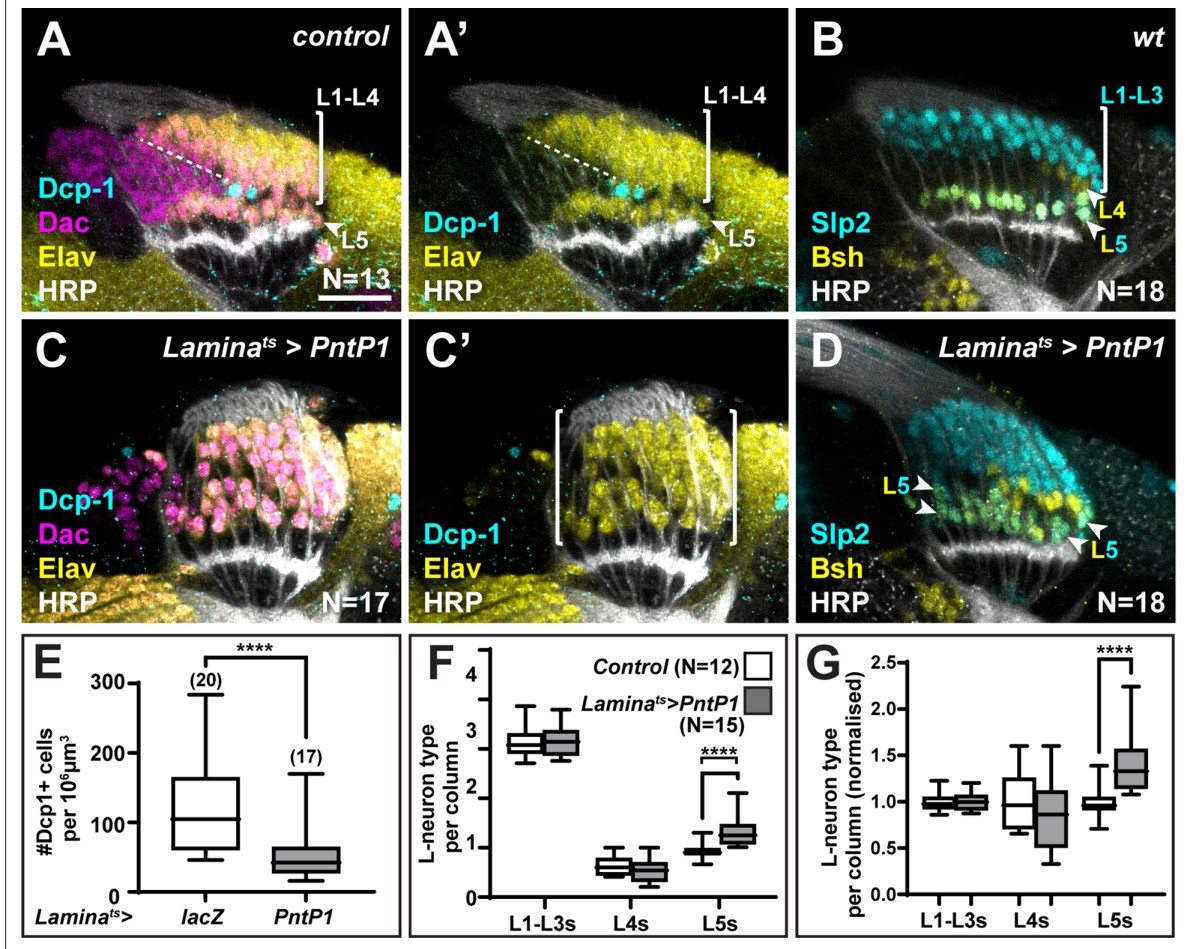

**Figure 4.** The 'extra' lamina precursor cells (LPCs) are specified as L5s. (**A**) Wild-type optic lobes stained for Dachshund (Dac) (magenta), Horseradish Peroxidase (HRP) (white), Embryonic lethal abnormal vision (Elav) (yellow), and cleaved Death caspase-1 (Dcp-1) (cyan). (**B**) Wild-type optic lobes stained for HRP (white) and L-neuron-type-specific markers sloppy paired 2 (Slp2) (cyan) and brain-specific homeobox (Bsh) (yellow). (**C, D**) Optic lobes with lamina-specific overexpression of PntP1 stained as in (**A**) and (**B**), respectively. (**C**) Fewer Dcp-1 positive cells were recovered compared with controls. (**D**) Roughly two rows of Slp2 and Bsh co-expressing cells (L5s) were recovered (arrowheads). (**E**) Quantification of the number of Dcp-1 positive cells in (**B**) compared with control *Lamina^ts>lacZ* (*Figure 4—figure supplement 1A*) (p<0.0001; Mann-Whitney U-test). (**F**) Quantification of the number of L-neuron types per column based on Slp2 and Bsh expression from column 7 onwards shows an increase in the number of L5s/column in *Lamina^ts>PntP1* compared with controls; p<0.0001; Mann-Whitney U-test. (**G**) Same as (**F**) but normalised to the mean of the control. The number of L5s/column in *Lamina^ts>PntP1* increase ~1.2-fold relative to controls; p<0.0001; Mann-Whitney U-test. Ns indicated in parentheses. Scale bar = 20 µm. For all quantifications boxes indicate the lower and upper quartiles; the whiskers represent the minimum and maximum values; the line inside the box indicates the median.

The online version of this article includes the following figure supplement(s) for figure 4:

**Figure supplement 1.** Hyperactivating Mitogen-activated protein kinase (MAPK) in the lamina drives ectopic L5 differentiation.

animals confirming that apoptosis was blocked (*Figure 5A*; N=26/26; with full penetrance). Indeed, we detected cells that were positive for the lamina marker Dachshund (Dac) but negative for the pan-neuronal marker Elav between L1-L4 and L5 neurons past column 5, which were never observed in controls (*Figure 5A* compared to *Figure 4A*; N=13/13; with full penetrance). These cells did not express lamina neuron-type markers Slp2 or Bsh, which L5s co-express and which individually label L1-L3s and L4s, respectively (*Figure 5B and C*). Thus, although the 'extra' LPCs were retained when apoptosis was blocked, they did not differentiate into neurons.

We observed ectopic L5s only when all LPCs were forced to differentiate, bypassing the need for differentiation signals from glia, but not when apoptosis was blocked (*Figure 4*, *Figure 4—figure supplement 1A-D*, and *Figure 5A–C*). This suggests that the 'extra' LPCs, though specified as L5s, did not receive differentiation signals from xg° in *Dronc* mutants or in the wild-type, where failing to

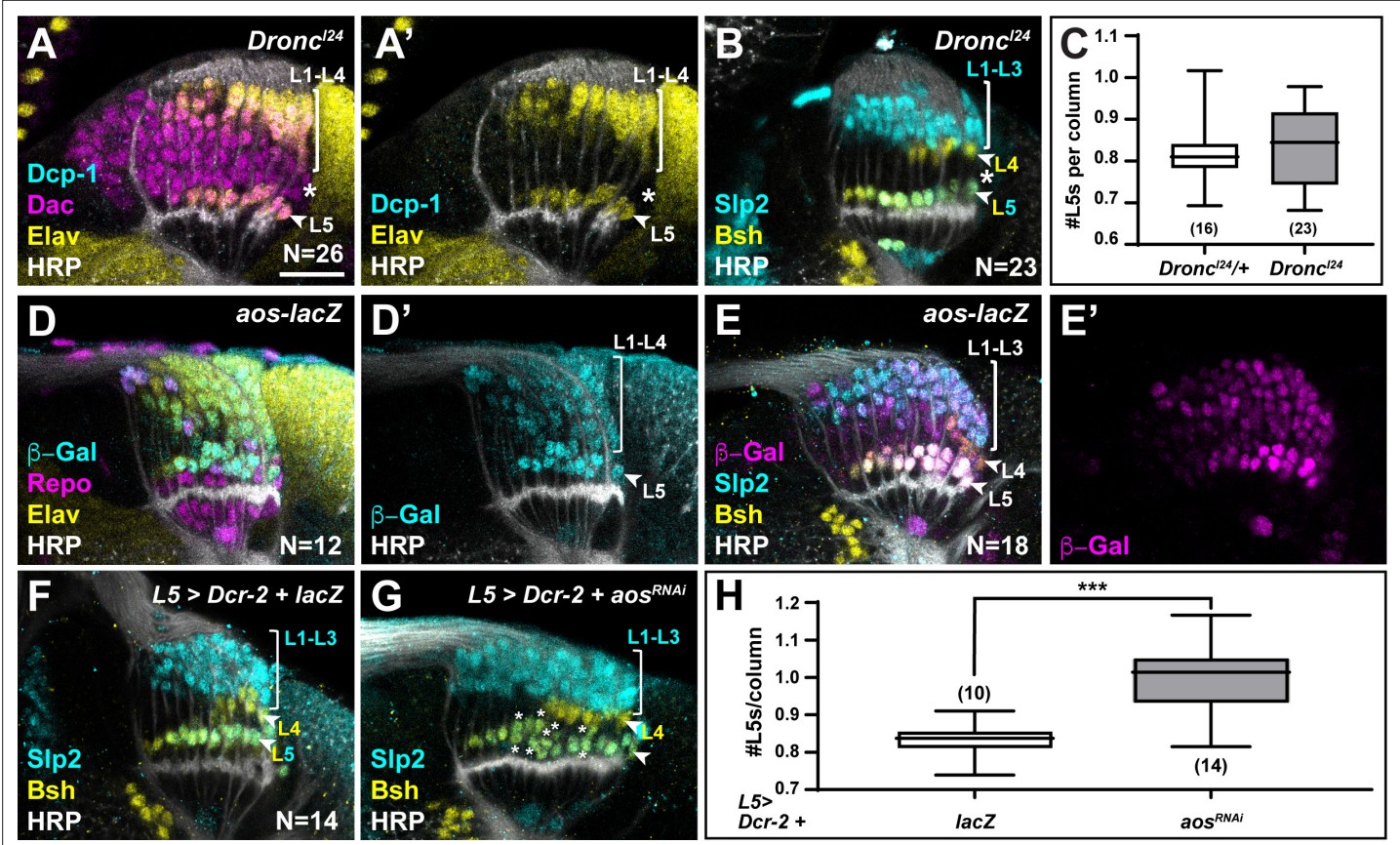

**Figure 5.** Newly induced L5 neurons secrete Aos to limit differentiation signals from xg[O]. (**A**) *Dronc[I24]* optic lobes stained for Death caspase-1 (Dcp-1) (cyan), Dachshund (Dac) (magenta), Embryonic lethal abnormal vision (Elav) (yellow), and Horseradish Peroxidase (HRP) (white). No Dcp-1 positive cells were recovered and Dac positive cells between L1-L4s and L5s persisted into the oldest columns (asterisk). (**B**) *Dronc[I24]* optic lobes stained for L-neuron-type-specific markers Sloppy paired 2 (Slp2) (cyan) and Brain-specific homeobox (Bsh) (yellow). A space (negative for both markers; asterisk) was present between L4s and L5s. (**C**) Quantifications for number of L5s/column in *Dronc[I24]* optic lobes compared to controls (*Dronc[I24]/+*) (p>0.05, Mann-Whitney U-test. Ns indicated in parentheses). (**D,E**) *aos-lacZ* expression in the lamina with (**D**) β-Galactosidase (β-Gal) (cyan), Repo (magenta), Elav (yellow), HRP (white), and with (**E**) β-Gal (magenta) and L-neuron-type-specific markers Slp2 (cyan), Bsh (yellow), as well as HRP (white). (**F**) An L5-specific Gal4 was used to drive expression of *Dcr-2* and *lacZ* in control lobes stained for Slp2 (cyan), Bsh (yellow), and HRP (white). (**G**) Optic lobes stained for HRP (white), Slp2 (cyan), and Bsh (yellow) when *Dcr-2* and *aos[RNAi]* were expressed in developing L5 neurons specifically, which led to an increase in the number of Slp2 and Bsh co-expressing cells (L5s; asterisks). (**H**) Quantification of the number of L5s/column for (**F**) and (**G**). ***p<0.0005; Mann-Whitney U-test. Ns indicated in parentheses. For all quantifications boxes indicate the lower and upper quartiles; the whiskers represent the minimum and maximum values; the line inside the box indicates the median. Scale bar = 20 μm.

The online version of this article includes the following figure supplement(s) for figure 5:

**Figure supplement 1.** Aos expression is delayed in younger L5s.

differentiate they were eliminated by apoptosis. How are only half of the LPCs specified as L5s chosen to differentiate in an invariant manner? The most proximal row of LPCs fated to differentiate into L5s is the row nearest to xg[O], and therefore, the first to receive differentiation signals. We speculated that newly induced L5s may then limit the ability of more distal LPCs to differentiate, by preventing MAPK activation in neighbouring cells. Aos is a transcriptional target of MAPK signalling and a secreted antagonist of the EGF Spi (*Freeman et al., 1992*; *Golembo et al., 1996*). We wondered if newly induced L5s secrete Aos to limit differentiation signals from xg[O]. To test this hypothesis, we examined *argos* (*aos*) expression with an enhancer trap in the *aos* locus, *aos[W11]*. *aos-lacZ* (*aos[W11]/+*) was expressed in xg[O] and differentiating lamina neurons, with the highest levels detected in L5s (*Figure 5D–E*). Interestingly, we also noted ectopic L5s in the laminas of *aos[W11]* heterozygotes, which could be the result of decreased Aos expression, as *aos[W11]* is a hypomorphic loss-of-function allele (*Figure 5E*). These observations suggested that Aos could act in L5s as a feedback-induced sink for Spi to limit further

differentiation in columns. To test this hypothesis, we knocked down *aos* by RNAi using a driver expressed specifically in developing L5s (*Jenett et al., 2012*; *Figure 5G*). We observed a statistically significant ~1.2-fold increase in the number of L5s relative to controls, that is, 0.99±0.02 L5s per column compared to 0.83±0.01 L5s per column in controls (*Figure 5F–H*; p<0.0005, Mann-Whitney U-test). Altogether, our data indicate a model in which xg<sup>O</sup> induce MAPK activity in the most proximal LPCs, resulting in their differentiation and in the production of the feedback inhibitor Aos. In turn, Aos limits further differentiation in the column by fine-tuning the availability of the differentiation signal Spi, which ensures that only one L5 differentiates per column, and determines the final number of neurons in each lamina column.

## Discussion

Appropriate circuit formation and function require that neuronal numbers are tightly regulated. This is particularly important for the visual system, which is composed of repeated modular circuits spanning multiple processing layers. In *Drosophila*, photoreceptors induce their target field, the lamina, thus, establishing retinotopy between the compound eye and the lamina (*Huang and Kunes, 1996*). Each lamina unit or column in the adult is composed of exactly five neurons; however, columns initially contain six LPCs. The sixth, or 'extra', LPC, invariantly located immediately distal to the differentiating L5 neuron, is fated to die by apoptosis. These 'extra' LPCs did not differentiate when apoptosis was blocked (*Figure 5A and B*) but generated ectopic L5s when forced to differentiate (*Figure 4D* and *Figure 4—figure supplement 1D*). Although we cannot rule out that preventing death using *Dronc* mutants may mis-specify the 'extra' cells and prevent them from differentiating, it is more likely that these 'extra' cells are specified as L5s, but that other mechanisms restricted their differentiation in *Dronc* mutants, as other lamina neuron types differentiated normally (*Figure 5B*). Thus, twice as many LPCs appear to be specified as L5s than undergo differentiation normally, which implies that a selection process to ensure that the correct number of L5s develop is in place.

The developmental strategies described thus far for setting neuronal number do so by regulating proliferation of precursors and/or survival of differentiated neurons (*Hidalgo and ffrench-Constant, 2003*). Here, we have defined a unique strategy whereby L5 neuronal numbers are set by regulating how many precursors from a larger pool are induced to differentiate, followed by programmed cell death of the excess precursors. We showed that a glial population called xg<sup>O</sup>, which are located proximal to the lamina, secrete at least two ligands (Spi, Col4a1) that activate MAPK signalling in LPCs to induce their differentiation (*Figure 3*, *Figure 3—figure supplement 1*). The tissue architecture is such that secreted signals from the xg<sup>O</sup> reach the most proximal row of LPCs first, and therefore these precursors differentiate first. Upon differentiation, these newly induced neurons secrete the Spi antagonist Aos to limit the available pool of Spi. As a result, the MAPK pathway is not activated in the 'extra' L5 LPCs, preventing them from differentiating into L5 neurons (*Figure 6*). Intriguingly, L5 neuronal differentiation in the youngest columns of the lamina proceeds despite Aos secretion by newly induced L5s. We noted that differentiating L5s expressed *aos* (based on *aos-lacZ*) at low levels initially and increased expression gradually till it plateaued from column 5 onwards (*Figure 5E* and *Figure 5—figure supplement 1B*). This delay in high *aos* expression may thus enable differentiation of the youngest LPCs, while still inhibiting differentiation of the row immediately distal. In sum, the structure of the tissue together with feedback from newly induced neurons set neuronal number by limiting which and, therefore, how many LPCs are induced to differentiate.

### Coordinating development through glia

We have shown that in addition to the wrapping glia (*Fernandes et al., 2017*), another population of glia, the xg<sup>O</sup>, also receive and relay signals from photoreceptors to induce neuronal differentiation in the lamina (*Figure 1E–F*). This is the first functional role ascribed to xg<sup>O</sup>. Remarkably, xg<sup>O</sup> are born from central brain DL1 type II neuroblasts and migrate into the optic lobes to positions below the developing lamina (*Ren et al., 2018*; *Viktorin et al., 2013*). This underscores an extraordinary degree of coordination and interdependence between the compound eye, optic lobe, and central brain. Photoreceptor signals drive wrapping glial morphogenesis and infiltration into the lamina (*Franzdóttir et al., 2009*), thus setting the pace of L1-L4 neuronal differentiation (*Fernandes et al.,*

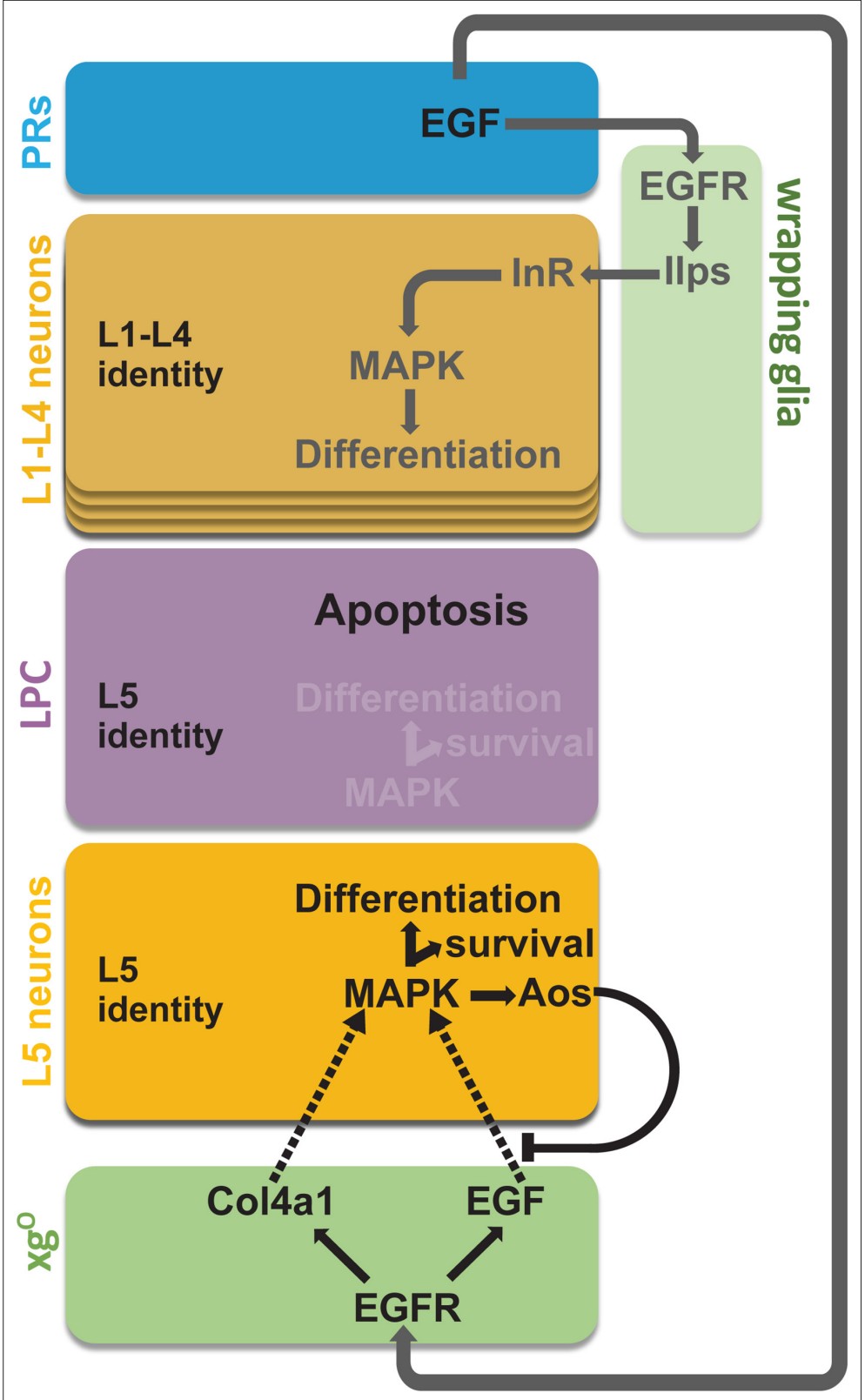

**Figure 6.** Summary schematic of neuronal differentiation in the lamina. In our model of lamina neuronal differentiation, lamina precursor cells (LPCs) are prepatterned with unique identities based on their positions within a column, such that the two most proximal cells are specified with L5 identity. Epidermal growth factor (EGF) from photoreceptors activates EGF receptor (EGFR) signalling in wrapping glia, which induce L1-L4 differentiation, and

*Figure 6 continued on next page*

*Figure 6 continued*

in xg$^O$, which induce L5 differentiation. Only a subset of the LPCs specified as L5s differentiate (i.e., those in the proximal row). We propose that this selective neuronal induction of L5s is due to tissue architecture and feedback from the newly born L5s, which limit available EGF (Spitz [Spi]) by secreting the antagonist Argos (Aos).

2017). Defining the signals that enable xg$^O$ to navigate the central brain and optic lobe will be a critical contribution to our understanding of how development is coordinated across brain regions.

## Tissue architecture sets up stereotyped programmed cell death

In both vertebrate and invertebrate developing nervous systems, programmed cell death is thought to come in two broad flavours: first as an intrinsically programmed fate whereby specific lineages or identifiable progenitors, neurons, or glia undergo stereotyped clearance (**Hidalgo and ffrench-Constant, 2003**; **Miguel-Aliaga and Thor, 2009**; **Pinto-Teixeira et al., 2016**; **Yamaguchi and Miura, 2015**) and second as an extrinsically controlled outcome of competition among neurons for limited target-derived trophic factors, which adjust overall cell numbers through stochastic clearance (also known as the neurotrophic theory) (**Davies, 2003**; **Hidalgo and ffrench-Constant, 2003**; **Miguel-Aliaga and Thor, 2009**; **Yamaguchi and Miura, 2015**). In the lamina, although the LPCs eliminated by programmed cell death are identifiable and the process stereotyped, it does not appear to be linked to an intrinsic programme. Rather, the predictable and stereotyped nature of apoptosis and differentiation are a consequence of stereotyped responses to extrinsic signalling determined by the architecture of the tissue. Thus, our work highlights that stereotyped patterns of apoptosis can arise from extrinsic signalling, suggesting a new mode to reliably pattern development of the nervous system.

In many contexts, neurotrophic factors promote cell survival by activating MAPK signalling (**Ballif and Blenis, 2001**; **Park and Poo, 2013**). In the lamina, MAPK-induced neuronal differentiation and cell survival appear intimately linked. LPCs that do not activate MAPK signalling sufficiently do not differentiate and are eliminated by apoptosis, likely through regulation of the proapoptotic factor Head involution defective, which has been described extensively in flies (**Bergmann et al., 2002**; **Bergmann et al., 1998**; **Kurada and White, 1998**). Thus, here the xg$^O$-secreted ligands Spi and Col4a1, which activate MAPK, appear to be functioning as differentiation signals as well as trophic factors. Col4a1, in particular, may perform dual roles by promoting MAPK activity directly through its receptor Ddr, and perhaps also by limiting Spi diffusivity to aid in localising MAPK activation.

It will be interesting to determine whether the processes described here represent conserved strategies for regulating neuronal number. Certainly, given the diversity of cell types and structural complexity of vertebrate nervous systems, exploiting tissue architecture would appear to be an effective and elegant strategy to regulate cell numbers reliably and precisely.

## Materials and methods

**Key resources table**

| Reagent type (species) or resource | Designation | Source or reference | Identifiers | Additional information |
|---|---|---|---|---|
| Genetic reagent (*Drosophila melanogaster*) | *Canton S* | Bloomington *Drosophila* Stock Center | BDSC: 64349 | |
| Genetic reagent (*Drosophila melanogaster*) | *Bacc-GFP* | Bloomington *Drosophila* Stock Center | BDSC: 36349 | |
| Genetic reagent (*Drosophila melanogaster*) | *ey-Gal80* | Bloomington *Drosophila* Stock Center | BDSC: 35822 | |
| Genetic reagent (*Drosophila melanogaster*) | *Gal80$^{ts}$* | Bloomington *Drosophila* Stock Center | BDSC: 7108 | |
| Genetic reagent (*Drosophila melanogaster*) | *Dronc$^{I24}$* | PMID:15800001 | | Gift from M Amoyel |
| Genetic reagent (*Drosophila melanogaster*) | *R27G05-Gal4* | Bloomington *Drosophila* Stock Center | BDSC: 48073 | Lamina Gal4 |

*Continued on next page*

*Continued*

| Reagent type (species) or resource | Designation | Source or reference | Identifiers | Additional information |
|---|---|---|---|---|
| Genetic reagent (*Drosophila melanogaster*) | R25A01-Gal4 | Bloomington *Drosophila* Stock Center | BDSC: 49102 | xg$^O$ Gal4 |
| Genetic reagent (*Drosophila melanogaster*) | R64B07-Gal4 | Bloomington *Drosophila* Stock Center | BDSC: 71106 | Larval L5 Gal4 |
| Genetic reagent (*Drosophila melanogaster*) | hh-gal4 | Bloomington *Drosophila* Stock Center | BDSC: 67493 | |
| Genetic reagent (*Drosophila melanogaster*) | Repo-Gal4 | Bloomington *Drosophila* Stock Center | BDSC: 7415 | |
| Genetic reagent (*Drosophila melanogaster*) | UAS-CD8::GFP | Bloomington *Drosophila* Stock Center | BDSC: 32187 | |
| Genetic reagent (*Drosophila melanogaster*) | UAS-nls.lacZ | Bloomington *Drosophila* Stock Center | BDSC: 3956 | |
| Genetic reagent (*Drosophila melanogaster*) | GMR-Gal4 | Bloomington *Drosophila* Stock Center | BDSC: 9146 | |
| Genetic reagent (*Drosophila melanogaster*) | Repo-QF | Bloomington *Drosophila* Stock Center | BDSC: 66477 | |
| Genetic reagent (*Drosophila melanogaster*) | NP6293-Gal4 | Kyoto Stock Center | DGRC: 105188 | Perineural Glia |
| Genetic reagent (*Drosophila melanogaster*) | NP2276-Gal4 | Kyoto Stock Center | DGRC: 112853 | Subperineur-al Glia |
| Genetic reagent (*Drosophila melanogaster*) | R54H02-Gal4 | Bloomington *Drosophila* Stock Center | BDSC: 45784 | Cortex Glia |
| Genetic reagent (*Drosophila melanogaster*) | R10C12-Gal4 | Bloomington *Drosophila* Stock Center | BDSC: 47841 | Epithelial and marginal glia |
| Genetic reagent (*Drosophila melanogaster*) | Mz97-Gal4 | Bloomington *Drosophila* Stock Center | BDSC: 9488 | Wrapping glia and xg$^O$ |
| Genetic reagent (*Drosophila melanogaster*) | R53H12-Gal4 | Bloomington *Drosophila* Stock Center | BDSC: 50456 | Chiasm glia |
| Genetic reagent (*Drosophila melanogaster*) | spi$^{NP0289}$-Gal4 | Kyoto Stock Center | DGRC: 112828 | |
| Genetic reagent (*Drosophila melanogaster*) | Cg-Gal4 | Bloomington *Drosophila* Stock Center | BDSC: 7011 | |
| Genetic reagent (*Drosophila melanogaster*) | bnl$^{NP2211}$-Gal4 | Kyoto Stock Center | DGRC: 112825 | |
| Genetic reagent (*Drosophila melanogaster*) | ths$^{MI07139}$-Gal4 | Bloomington *Drosophila* Stock Center | BDSC: 77475 | |
| Genetic reagent (*Drosophila melanogaster*) | rho3$^{PLLb}$, UAS-CD8::GFP | PMID:20957186 | | Gift from B Shilo |
| Genetic reagent (*Drosophila melanogaster*) | UAS-rho3-3xHA | PMID:20957186 | | Gift from B Shilo |
| Genetic reagent (*Drosophila melanogaster*) | aos$^{w11}$ | Bloomington *Drosophila* Stock Center | BDSC: 2513 | aos-lacZ |
| Genetic reagent (*Drosophila melanogaster*) | BaccGFP;10xQUAS-6xmCherry-HA | Bloomington *Drosophila* Stock Center | BDSC: 55270 | |
| Genetic reagent (*Drosophila melanogaster*) | 10xUAS-myrGFP | Bloomington *Drosophila* Stock Center | BDSC: 32197 | |
| Genetic reagent (*Drosophila melanogaster*) | UAS-LifeAct-GFP | Bloomington *Drosophila* Stock Center | BDSC: 35544 | |

*Continued on next page*

*Continued*

| Reagent type (species) or resource | Designation | Source or reference | Identifiers | Additional information |
|---|---|---|---|---|
| Genetic reagent (*Drosophila melanogaster*) | UAS-Dicer2 | Bloomington *Drosophila* Stock Center | BDSC: 24650 | |
| Genetic reagent (*Drosophila melanogaster*) | ;UAS-EGFR$^{DN}$; UAS-EGFR$^{DN}$ | Bloomington *Drosophila* Stock Center | BDSC: 5364 | |
| Genetic reagent (*Drosophila melanogaster*) | UAS-rl$^{SEM}$ | Bloomington *Drosophila* Stock Center | BDSC: 59006 | rlS$^{EM}$ = MAPK$^{ACT}$ |
| Genetic reagent (*Drosophila melanogaster*) | UAS-PntP1 | Bloomington *Drosophila* Stock Center | BDSC: 869 | |
| Genetic reagent (*Drosophila melanogaster*) | UAS-jeb | PMID:21816278 | | Gift from A Gould |
| Genetic reagent (*Drosophila melanogaster*) | UAS-Col4a1$^{EY11094}$ | Bloomington *Drosophila* Stock Center | BDSC: 20661 | |
| Genetic reagent (*Drosophila melanogaster*) | UAS-Cg25cRFP | PMID:26090908 | | Gift from A Franz Cg25c=Col4a1 |
| Genetic reagent (*Drosophila melanogaster*) | UAS-wnt5 | Bloomington *Drosophila* Stock Center | BDSC: 64298 | |
| Genetic reagent (*Drosophila melanogaster*) | UAS-s.spi | PMID:7601354 | | Gift from B Shilo |
| Genetic reagent (*Drosophila melanogaster*) | UAS-m.spi::GFP-myc (II) | PMID:11799065 | | Gift from B Shilo m.spi=spi$^{wt}$ |
| Genetic reagent (*Drosophila melanogaster*) | UAS-m.spi::GFP-myc (III) | PMID:11799065 | | Gift from B Shilo m.spi=spi$^{wt}$ |
| Genetic reagent (*Drosophila melanogaster*) | UAS-grk.sec | Bloomington *Drosophila* Stock Center | BDSC: 58417 | |
| Genetic reagent (*Drosophila melanogaster*) | UAS-vn$^{EPgy}$ | Bloomington *Drosophila* Stock Center | BDSC: 58498 | |
| Genetic reagent (*Drosophila melanogaster*) | UAS-krn-3xHA | FlyORF | F002754 | |
| Genetic reagent (*Drosophila melanogaster*) | UAS-bnl | Bloomington *Drosophila* Stock Center | BDSC: 64232 | |
| Genetic reagent (*Drosophila melanogaster*) | UAS-Ilp1 | PMID:12176357 | | Gift from P Leopold |
| Genetic reagent (*Drosophila melanogaster*) | UAS-Ilp6 | PMID:20059956 | | Gift from P Leopold |
| Genetic reagent (*Drosophila melanogaster*) | UAS-Pvf1$^{XP}$ | Bloomington *Drosophila* Stock Center | BDSC: 19632 | |
| Genetic reagent (*Drosophila melanogaster*) | UAS-Pvf2$^{XP}$ | Bloomington *Drosophila* Stock Center | BDSC: 19631 | |
| Genetic reagent (*Drosophila melanogaster*) | UAS-Wnt4$^{EPgy2}$ | Bloomington *Drosophila* Stock Center | BDSC: 20162 | |
| Genetic reagent (*Drosophila melanogaster*) | UAS-boss-3xHA | FlyORF | F001365 | |
| Genetic reagent (*Drosophila melanogaster*) | SAM.dCas9.Trk | Bloomington *Drosophila* Stock Center | BDSC: 81322 | |
| Genetic reagent (*Drosophila melanogaster*) | SAM.dCas9.Pvf3 | Bloomington *Drosophila* Stock Center | BDSC: 81346 | |
| Genetic reagent (*Drosophila melanogaster*) | SAM.dCas9.ths | Bloomington *Drosophila* Stock Center | BDSC: 81347 | |

*Continued*

| Reagent type (species) or resource | Designation | Source or reference | Identifiers | Additional information |
|---|---|---|---|---|
| Genetic reagent (*Drosophila melanogaster*) | *SAM.dCas9.pyr* | Bloomington *Drosophila* Stock Center | BDSC: 81330 | |
| Genetic reagent (*Drosophila melanogaster*) | *Ddr^CR01018^-Gal4* | Bloomington *Drosophila* Stock Center | BDSC: 81157 | |
| Genetic reagent (*Drosophila melanogaster*) | *spi^RNAi^* | Vienna *Drosophila* Stock Center | GD3922 | |
| Genetic reagent (*Drosophila melanogaster*) | *Col4a1^RNAi^* | Vienna *Drosophila* Stock Center | GD28369 | |
| Genetic reagent (*Drosophila melanogaster*) | *aos^RNAi^* | Vienna *Drosophila* Stock Center | GD47181 | |
| Antibody | Anti-Dac2-3 (mouse monoclonal) | Developmental Studies Hybridoma Bank | mAbdac2-3 | 1:20 |
| Antibody | Anti-Repo (mouse monoclonal) | Developmental Studies Hybridoma Bank | 8D12 | 1:20 |
| Antibody | Anti-Elav (rat monoclonal) | Developmental Studies Hybridoma Bank | 7E8A10 | 1:100 |
| Antibody | Anti-Elav (mouse monoclonal) | Developmental Studies Hybridoma Bank | 9F8A9 | 1:20 |
| Antibody | Anti-Svp (mouse monoclonal) | Developmental Studies Hybridoma Bank | 6F7 | 1:50 |
| Antibody | Anti-Slp2 (guinea pig polyclonal) | PMID:23783517 | C Desplan | 1:100 |
| Antibody | Anti-Bsh (Rabbit polyclonal) | PMID:33149298 | C Desplan | 1:500 |
| Antibody | Anti-Dcp-1 (Rabbit polyclonal) | Cell Signaling | 9578 | 1:100 |
| Antibody | Anti-Brp (guinea pig polyclonal) | | C Desplan | 1:100 |
| Antibody | Anti-phospho-p44/42-MAPK (Thr202/Tyr204) (Rabbit polyclonal) | Cell Signaling | 9101 | 1:100 |
| Antibody | Anti-β-galactosidase (mouse monoclonal) | Promega | #Z3781 | 1:500 |
| Antibody | Anti-β-galactosidase (chicken polyclonal) | abcam | 9361 | 1:500 |
| Antibody | Anti-GFP (chicken polyclonal) | EMD Millipore | GFP-1010 | 1:400 |
| Antibody | Anti-Pdm3 (rat polyclonal) | PMID:22190420 | C Desplan | 1:20 |
| Antibody | Anti-RFP (chicken polyclonal) | Rockland | #600-901-379s | 1:500 |
| Antibody | Anti-GFP (rabbit polyclonal) | Thermo Fisher Scientific | #A6455 | 1:500 |
| Antibody | AlexaFluor405-conjugated Goat Anti-HRP (goat polyclonal) | Jackson Immunolabs | 123-475-021 | 1:200 |
| Antibody | AlexaFluorCy3- conjugated Goat Anti-HRP (goat polyclonal) | Jackson Immunolabs | 11 23-165-021 | 1:200 |
| Antibody | AlexaFluor647- conjugated Goat Anti-HRP (goat polyclonal) | Jackson Immunolabs | 123-605-021 | 1:200 |

*Continued on next page*

Continued

| Reagent type (species) or resource | Designation | Source or reference | Identifiers | Additional information |
|---|---|---|---|---|
| Sequence-based reagent | Antisense probe pairs for *in situ* Hybridisation chain reaction | This study. 'Prasad et al. HCR Probe Sequences.xls' | DNA Oligos | *Figure 3—source data 1* |
| Software, algorithm | RStudio | RStudio | R version 4.0.3 | |
| Software, algorithm | GraphPad Prism 9 | GraphPad Prism 9 | GraphPad Prism version 9.4.1 | |
| Software, algorithm | Adobe Photoshop | Adobe Photoshop | Adobe Photoshop 2021 | |
| Software, algorithm | Adobe Illustrator | Adobe Illustrator | Adobe Illustrator 2021 | |
| Software, algorithm | Imaris | Imaris | Imaris ×64-9.5.1 | |
| Software, algorithm | FiJi, ImageJ | PMID:22743772 | | |
| Chemical compound, drug | HCR Amplification Buffer | Molecular Instruments | BAM02224 | |
| Chemical compound, drug | HCR Wash Buffer | Molecular Instruments | BPW02124 | |
| Chemical compound, drug | HCR Hybridisation Buffer | Molecular Instruments | BPH02224 | |
| Chemical compound, drug | HCR Amplifier B3-H1-546 | Molecular Instruments | S030724 | |
| Chemical compound, drug | HCR Amplifier B3-H2-546 | Molecular Instruments | S031024 | |
| Chemical compound, drug | HCR Amplifier B3-H1-647 | Molecular Instruments | S040124 | |
| Chemical compound, drug | HCR Amplifier B3-H2-647 | Molecular Instruments | S040224 | |
| Chemical compound, drug | Para-formaldehyde | Thermo Fisher Scientific | 28908 | 4% solution |
| Chemical compound, drug | DAPI stain | Sigma | D9542-1MG | (1 µg/mL) |

## *Drosophila* stocks and maintenance

*Drosophila melanogaster* strains and crosses were reared on standard cornmeal medium and raised at 25°C or 29°C or shifted from 18°C to 29°C for genotypes with temperature-sensitive Gal80, as indicated in *Supplementary file 2*.

We used the following mutant and transgenic flies in combination or recombined in this study (see Supporting File 2 for more details; {} enclose individual genotypes, separated by commas).

{y,w,hsflp¹²²; sp/Cyo; TM2/TM6B}, {y,w; sp/Cyo, Bacc-GFP; Dr/TM6C}, (from BDSC: 36349).

{ey-Gal80; sp/Cyo;} (BDSC: 35822), {;Gal80ᵗˢ; TM2/TM6B} (BDSC: 7108), {w¹¹¹⁸;; R27G05-Gal4} (BDSC: 48073), {w¹¹¹⁸;;25A01-Gal4} (BDSC: 49102), {y,w; R64B07-Gal4;} (larval L5-Gal4), {y,w; hh-Gal4/TM3} (BDSC: 67493), {;tub-Gal80ᵗˢ; repo-Gal4/TM6B}, {w¹¹¹⁸;GMR-Gal4/Cyo;} (BDSC: 9146), {y,w;Pin/Cyo;repo-QF/TM6B} (BDSC: 66477), {y,w; NP6293-Gal4/Cyo,UAS-lacZ;} (perineurial glia; Kyoto Stock Center: 105188), {w; NP2276-Gal4/Cyo; } (subperineurial glia; Kyoto Stock Center: 112853), {w¹¹¹⁸;; R54H02-Gal4} (cortex glia; BDSC: 45784), {w¹¹¹⁸;; R10C12-Gal4} (epithelial and marginal glia; BDSC: 47841), {w;Mz97-Gal4, UAS-Stinger/Cyo;} (wrapping and xgᴼ; BDSC: 9488), {w¹¹¹⁸;; R53H12-Gal4} (chiasm glia; BDSC: 50456), {y,w; spiᴺᴾ⁰²⁸⁹-Gal4/Cyo, UAS-lacZ;} (Kyoto Stock Center: 112128), {w¹¹¹⁸; Cg-Gal4;} (BDSC: 7011), {w;; bnlᴺᴾ²²¹¹-Gal4} (Kyoto Stock Center: 112825), {w; thsᴹᴵ⁰⁷¹³⁹-Gal4/Cyo; MKRS/TM6B} (BDSC: 77475), {;;rho3ᴾᴸᴸᵇ, UAS-CD8::GFP/TM6B}, {;UAS-rho3-3xHA;} (gifts from B Shilo), {;;aosʷ¹¹/TM6B} (aos-lacZ; BDSC: 2513), {y,w; sp/Cyo, Bacc-GFP; 10xQUAS-6xmCherry-HA} (BDSC: 52270), {y,w;;10xUAS-myrGFP} (BDSC: 32197), {;UAS-CD8::GFP;}, {;;UAS-CD8::GFP} (gifts from C Desplan), {y,w;;UAS-nls.lacZ}, (BDSC: 3956), {y,w; UAS-LifeAct-GFP/Cyo;} (BDSC: 35544), {w¹¹¹⁸;UAS-Dcr-2;} (BDSC: 24650), {w¹¹¹⁸;;UAS-Dcr-2} (BDSC: 24651), {;UAS-EGFRᴰᴺ; UAS-EGFRᴰᴺ} (BDSC: 5364), {;UAS-aopᴬᶜᵀ;} (Kyoto Stock Center: 108425), {y,w;UAS-rlˢᵉᵐ;} (rlˢᵉᵐ = MAPKᴬᶜᵀ; BDSC: 59006), {w¹¹¹⁸;;UAS-PntP1} (BDSC: 869), {w¹¹¹⁸;UAS-aosᴿᴺᴬⁱ;} (VDRC47181), {w;UAS-jeb;} (a gift from A Gould), {y,w, UAS-Col4a1ᴱʸ¹¹⁰⁹⁴/(Cyo);} (BDSC: 20661), {;;UAS-Cg25c-RFP} (*Zang et al., 2015*) (Col4a1=Cg25c), {;UAS-Wnt5;} (BDSC: 64298), {;;UAS-s.spi} (a gift from B Shilo), {;;UAS-m.spi::GFP-myc;} (a gift from B Shilo), {;;UAS-m.spi::GFP-myc} (a gift from B Shilo), {w, UAS-grk.sec/Cyo;} (BDSC: 58417), {;UAS-vnᴱᴾᵍʸ/Cyo;} (BDSC: 58498), {;;UAS-krn-3xHA} (FlyORF: F002754), {;UAS-bnl/Cyo;

*MKRS/TM6C}* (BDSC: 64232), *{;UAS-Ilp1;}*, *{;UAS-Ilp6;}* (gifts from P Leopold), *{w^{1118}, UAS-Pvf1^{XP};;}* (BDSC: 19632), *{w^{1118}; UAS-Pvf2^{XP};}* (BDSC: 19631), *{;UAS-Wnt4^{EPgy2}/Cyo;}* (BDSC: 20162), *{;;UAS-boss-3xHA}* (FlyORF: F001365), *{y,sev; SAM.dCas9.Trk;}* (BDSC: 81322), *{y,sev; SAM.dCas9.Pvf3;}* (BDSC: 81346), *{y,sev; SAM.dCas9.ths;}* (BDSC: 81347), *{y,sev; SAM.dCas9.pyr;}* (BDSC: 81330), *{w^{1118}; Ddr^{CR01018}-Gal4;}* (BDSC: 81157).

## Immunocytochemistry, antibodies, and microscopy

We dissected eye-optic lobe complexes from early pupae (0–5 hr after puparium formation) in ×1 phosphate-buffered saline (PBS), fixed in 4% formaldehyde for 20 min, blocked in 5% normal donkey serum, and incubated in primary antibodies diluted in block for two nights at 4°C. Samples were then washed in ×1 PBS with 0.5% Triton-X (PBSTx), incubated in secondary antibodies diluted in block, washed in PBSTx and mounted in SlowFade (Life Technologies).

When performing phospho-MAPK stains, dissections were performed in a phosphatase inhibitor buffer as detailed in *Amoyel et al., 2016*.

We used the following primary antibodies in this study: mouse anti-Dac[2-3] (1:20, Developmental Studies Hybridoma Bank [DSHB]), mouse anti-Repo (1:20, DSHB), rat anti-Elav (1:100, DSHB), mouse anti-Elav (1:20, DSHB), rabbit anti-Dcp-1 (1:100; Cell Signalling #9578), chicken anti-GFP (1:400; EMD Millipore), mouse anti-Svp (1:50, DSHB), rabbit anti-Slp2 (1:100; a gift from C Desplan), rabbit-Bsh (1:500; a gift from C Desplan), Rat anti-Pdm3 (1:1000; a gift from C Desplan), guinea pig anti-Brp (1:100; a gift from C Desplan), rabbit anti-Phospho-p44/42 MAPK (Erk1/2) (Thr202/Tyr204) (1:100, Cell Signaling #9101), chicken anti-RFP (1:500; Rockland #600-901-379s), mouse anti-β-galactosidase (1:500; Promega #Z3781), chicken anti-β-galactosidase (1:500; abcam #9361), rabbit-anti-GFP (1:500; Thermo Fisher Scientific #A6455), AlexaFluor405 conjugated Goat Anti-HRP (1:100; Jackson Immunolabs), AlexaFluor405-, Cy3-, or AlexaFluor647-conjugated Goat Anti-HRP (1:200; Jackson Immunolabs). Secondary antibodies were obtained from Jackson Immunolabs or Invitrogen and used at 1:800. Images were acquired using Zeiss 800 and 880 confocal microscopes with ×40 objectives.

## In situ hybridisation chain reaction

To determine if *spi, Col4a1, and Ddr* transcripts were present in the xg^O, we performed HCR as detailed in *Duckhorn et al., 2022*. We designed 20–21 probe pairs against target genes, excluding regions of strong similarity to other transcripts, with corresponding initiator sequences for amplifiers B3 (*Choi et al., 2018*). HCR probes (sequences included as source data; see *Figure 3—source data 1*) were purchased as DNA Oligos from Thermo Fisher Scientific (100 µm in water and frozen).

Eye-optic lobe complexes were dissected, fixed, and washed as detailed above. Samples were incubated in Probe Hybridisation Buffer for 30 min at 37°C followed by incubation with probes (0.01 µM) at 37°C overnight. The samples were then washed four times for 15 min each with probe wash buffer at 37°C followed by two washes for 5 min each with ×5 saline sodium citrate solution (20XSSCT solution in distilled water – 58.44 g/mol sodium chloride, 294.10 g/mol 560 sodium citrate, pH adjusted to 7 with 14 N hydrochloric acid, with 0.001% Tween 20) at room temperature. Samples were then incubated with amplification buffer for 10 min at room temperature. 12 pmol of hairpins H1 and H2 were snap-cooled (95°C for 90 s and then cooled to room temperature for 20 min) separately to avoid oligomerisation. The snap-cooled hairpins were then added to the samples in the amplification buffer (protected from light) and incubated overnight at room temperature. The samples were then washed with 5XSSCT for 15 min before being incubated in darkness with 1:15 dilution of DAPI (Sigma D9542) for 90 min. Samples were washed with ×1 PBS for 30 min and then mounted as detailed above.

## Quantification and statistical analyses

We used Fiji-ImageJ (*Schindelin et al., 2012*) or Imaris (version x64-9.5.1) to process and quantify confocal images as described below. We used Adobe Photoshop and Adobe Illustrator software to prepare figures. We used GraphPad Prism 8 to perform statistical tests. In all graphs, whiskers indicate the standard error of the mean (SEM).

## Dcp-1 quantifications

We used the surfaces tool in Imaris to manually select the lamina region (based on Dac expression). We then used the spots tool to identify Dcp-1 positive cells (cell diameter = 5 µm) within the selected

region using the default thresholding settings, and plotted these values normalised to the volume of the selected lamina region in GraphPad Prism 8.

## Cell-type quantifications

### LPCs per column

Column number was identified by counting HRP-labelled photoreceptor axon bundles. We considered the youngest column located adjacent to the lamina furrow to be the first column, with column number (age) increasing towards the posterior (right) of the furrow. We counted the number of Dac+ cells per column by quantifying 10 optical slices (step size = 1 μm) located centrally in the lamina.

### *Control vs. Lamina*[ts]*>PntP1*

We quantified the lamina neuron types per column using the following markers to identify L-neuron types: Elav+ and Slp2+ cells were counted as L1-L3s; Elav+ and Bsh+ cells were counted as L4s and Elav+, Bsh+, and Slp2+ cells were counted as L5s. We quantified 10 optical slices (step size = 1 μm) located centrally in the lamina. Column number was identified by counting HRP-labelled photoreceptor axon bundles. These quantifications were done blind.

### Ligand receptor screen

We quantified the number of L5s based on Elav expression in the proximal lamina. Column number was identified by counting HRP-labelled photoreceptor axon bundles. We quantified 30 optical slices (step size = 1 μm) located centrally in the lamina.

## Ligand overexpression quantifications

We quantified the number of L-neuron types per column using Elav, Bsh, and Slp2. We quantified 30 optical slices (step size = 1 μm) located centrally in the lamina. Column number was identified by counting HRP-labelled photoreceptor axon bundles.

## Spi and Col4a1 probe intensity quantifications

In Fiji-ImageJ we used the free hand selection tool to draw a region of interest (ROI) around the xg$^O$ (marked by the xg$^O$>CD8::GFP). We then measured the mean fluorescence intensity (MFI) of *spi* and *Col4a1* transcripts labelled by HCR within each ROI. We quantified 30 optical slices (step size = 1 μm) located centrally in the lamina and then plotted the average for each optic lobe.

## Number of xg$^O$

We quantified the number of xg$^O$ (***Figure 1—figure supplement 1Q***) by manually counting the number of Repo positive nuclei within LifeAct-GFP positive xg$^O$ per 40 μm optical section in Fiji-ImageJ. We used a step size of 1 μm while acquiring the z-stacks and centred each 40 μm optical section in the middle of the lamina using photoreceptor axons (HRP), and the lobula plug (Dac expression) as landmarks. Quantifications were performed blind.

## Length of xg$^O$ processes

We quantified the lengths of the fine glial processes that extend distally from the xg$^O$ towards the lamina plexus (***Figure 1—figure supplement 1O,P,R***) by using the straight-line selection and measuring tools in Fiji-ImageJ to measure xg$^O$ process lengths in a 10 μm optical section centred in the middle of the lamina. Quantifications were performed blind.

## Quantifications of nuclear to cytoplasmic dpMAPK MFI

Using Fiji we manually drew ROIs with the free hand selection tool around the xg$^O$ nucleus (based on Repo) and LPCs in the most proximal row of the lamina (based on Dac expression) and added these to the ROI manager. We then enlarged the ROIs (Edit > Selection > Enlarge) by 3.00 pixel units to include the cytoplasm. We then used the XOR function in the ROI Manager to only select the cytoplasm of the xg$^O$. We then measured the MFI of dpMAPK in the nucleus and the cytoplasm of the xg$^O$ in 20 centrally located optical slices (corresponding to 20 μm) for each optic lobe. We plotted the nuclear:-cytoplasmic ratios of dpMAPK MFI in GraphPad Prism 8.

## aos-lacZ intensity quantifications

Using Fiji we manually drew regions of interest around L5s (based on Slp2+Bsh co-expression) in each column. We then measured the MFI of β-Galactosidase in the ROIs. We quantified 10 optical slices (step size = 1 μm) for each optic lobe and plotted the average values as a function of column (age).

## Acknowledgements

We thank C Desplan, A Gould, B Shilo, and J Treisman for reagents, and S Ackerman, M Amoyel, B Conradt, C Desplan, C Doe, A Franz, P Salinas, A Rossi, C Stern, L Venkatasubramanian, and members of the Amoyel and Fernandes labs for comments on the manuscript. Stocks obtained from the Bloomington Drosophila Stock Center (NIH P40OD018537) were used in this study. Monoclonal antibodies obtained from the Developmental Studies Hybridoma Bank, created by the NICHD of the NIH and maintained at The University of Iowa, were used in this study. Funding: Wellcome Trust Sir Henry Dale Research Fellowship 210472/Z/18/Z (VMF), UCL Overseas Research Scholarship (ARP), and UCL Graduate Research Scholarship (ARP).

## Additional information

### Competing interests

Vilaiwan M Fernandes: Reviewing editor, eLife. The other authors declare that no competing interests exist.

### Funding

| Funder | Grant reference number | Author |
|---|---|---|
| Wellcome Trust | 210472/Z/18/Z | Vilaiwan M Fernandes |
| UCL Overseas Research Scholarship | | Anadika R Prasad |
| UCL Graduate Research Scholarship | | Anadika R Prasad |
| UCL Biosciences Graduate Research Scholarship | | Matthew P Bostock |

The funders had no role in study design, data collection and interpretation, or the decision to submit the work for publication. For the purpose of Open Access, the authors have applied a CC BY public copyright license to any Author Accepted Manuscript version arising from this submission.

### Author contributions

Anadika R Prasad, Conceptualization, Formal analysis, Validation, Investigation, Visualization, Methodology, Writing – original draft, Writing – review and editing; Inês Lago-Baldaia, Formal analysis, Investigation, Visualization, Methodology, Writing – review and editing; Matthew P Bostock, Zaynab Housseini, Investigation; Vilaiwan M Fernandes, Conceptualization, Supervision, Funding acquisition, Validation, Investigation, Visualization, Methodology, Writing – original draft, Project administration, Writing – review and editing

### Author ORCIDs

Anadika R Prasad http://orcid.org/0000-0003-4067-1784
Vilaiwan M Fernandes http://orcid.org/0000-0002-1991-7252

### Decision letter and Author response

Decision letter https://doi.org/10.7554/eLife.78092.sa1
Author response https://doi.org/10.7554/eLife.78092.sa2

## Additional files

### Supplementary files
• Supplementary file 1. Table summarising the results from the glial-Gal4 screen (*Figure 1B and C*, *Figure 1—figure supplement 1B-N*) to identify the glial type that regulates L5 development.

• Supplementary file 2. Table listing all genotypes and experimental conditions used by figure panel. (Note that only female genotypes are listed through both sexes were included in our analyses.)

• Transparent reporting form

### Data availability
All data generated or analysed during this study are included in the manuscript.

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
