## [Editor Report]

This manuscript describes how control over the induction of neuronal fate from a progenitor pool regulates the generation of the appropriate numbers of neurons in the developing *Drosophila* retina. It describes how this occurs non-autonomously through two distinct glial cell types. It will be of interest to cell and developmental biologists and neuroscientists.

---

## [Decision Letter]

**Decision letter after peer review:**

Thank you for submitting your article "Differentiation signals from glia are fined-tuned to set neuronal numbers during development" for consideration by *eLife*. Your article has been reviewed by 3 peer reviewers, including Sonia Sen as Reviewing Editor and Reviewer #1, and the evaluation has been overseen by a Reviewing Editor and K VijayRaghavan as the Senior Editor.

Essential revisions:

1. Loss of function of Spi and Col4a1: In the absence of loss-of-function data for these ligands one can't rule out the possibility that their source is the photoreceptors. Could the authors please demonstrate that the outer chiasm glia is indeed their source by performing loss-of-function analysis within the glia?

2. Statistics: Could the authors please check their statistics? Non-parametric tests are used throughout the manuscript – are their data not normally distributed? We also recommend that the authors use graphs that show the distribution of their data. In the detailed reviews below are more specific comments regarding the statistics. Could the authors please pay attention to each of these points as they revise their manuscript?

3. Enhancer trap lines: We were concerned about the validity of the enhancer trap lines. If any of them have been demonstrated in earlier studies to be faithful, could the authors please state that explicitly? For any lines that haven't been verified, can the authors please discuss the caveats also explicitly?

4. Spi mRNA in situ: The Spi mRNA in situ is ambiguous and removing it will not change the storyline. We recommend removing this data. Alternatively, the authors would need to address the concerns raised by the reviewers.

5. Overlap with Fernandes et al., 2017: The current manuscript rests heavily on the previous one. Many of the manipulations used in some of these assays were described earlier. The manner in which the manuscript is currently written may be laying equal emphasis on the previous discoveries and the novelty in this current one. For example, the screen to discover which EGFR ligand might be involved is described in two action-packed sentences : We suggest that the authors unpack and emphasise the novel aspects of *this* story ( xgo > L5 neuron specification > inhibition of the second set of L5s) to avoid giving the (false) impression that it is an overlap of their earlier one.

6. The 'un-dead' L5s: There is no guarantee that the ectopic lamina precursor cells found in Dronc mutant are identical to the cells that fail to differentiate to L5 due to insufficient MAPK signalling. The forced MAPK activation in Figure 1H, I does not clarify this point. Could the authors please revisit the discussion around these precursors to highlight this point?

7. Quantification of nuclear MAPKinase: It is not clear where dpMAPKinase is in the L5 neurons in Figure S3P, Q (nuclear or not). (The quantification in S3L is for xgo glia.) So, in the absence of quantification, it isn't clear how the authors can tell in which cells dpMAPkinase signal is. We recommend quantifying dMAPK in L5 in the two scenarios where Spi and Col4a are over-expressed in the xgo glia.

---

## [Author Response]

Essential revisions:1. Loss of function of Spi and Col4a1: In the absence of loss-of-function data for these ligands one can't rule out the possibility that their source is the photoreceptors. Could the authors please demonstrate that the outer chiasm glia is indeed their source by performing loss-of-function analysis within the glia?

We thank the reviewers for suggesting these experiments. To test whether Spi and Col4a1 specifically from outer chiasm giant glia (xg^o^), and not photoreceptors, induce L5 neuronal differentiation, we used previously validated RNAi lines to knock down *spi* and *Col4a1* (Chen et al., 2016; Csordás et al., 2020; Louradour et al., 2017; Morante et al., 2013; PastorPareja and Xu, 2011), individually and simultaneously in the xg^O^. Knocking down *Col4a1* specifically in xg^O^ led to a statistically significant decrease in the number of L5s per column (~15.2% decrease relative to controls; one-way ANOVA with Dunn’s multiple comparisons test; P<0.05), while knocking *spi* produced a milder decrease in the number of L5s per column (~11.5% decrease relative to controls), albeit this was not statistically significant with an one-way ANOVA with Dunn’s multiple comparisons test, but was statistically significant with a pair-wise comparison by Mann-Whitney U-test (P<0.0005). However, and most importantly, knocking down both *spi* and *Col4a1* simultaneously in xg^O^, led to a strong and statistically significant decrease in the number of L5s per column (~33% decrease relative to controls). Thus, altogether our data suggest that Spi and Col4a1 specifically from xg^O^ induce L5 neuronal differentiation. We have updated the text and Figures to include these data as follows:

Page 9, lines 252-268 (and Figure 3V-W):

“Next, to test whether xg^O^-derived Spi and Col4a1 are normally required to induce L5 neuronal differentiation, we disrupted their expression specifically in xg^O^. We used RNA interference (RNAi) to knock down *spi* and *Col4a1* expression both individually and simultaneously in xg^O^ using previously validated lines (Chen et al., 2016; Csordás et al., 2020; Louradour et al., 2017; Morante et al., 2013; Pastor-Pareja and Xu, 2011). While knocking down *spi* led to a mild decrease in L5 numbers, which was not statistically significant, knocking down *Col4a1* in the xg^O^ led to a statistically significant decrease in L5s (0.78 ± 0.03 L5s per column) relative to controls (0.92 ± 0.02 L5s per column) (Figure 3T, 3U, 3W; P*<0.05 one-way ANOVA with Dunn’s multiple comparisons test). However, knocking down both *spi* and *Col4a1* simultaneously in xg^O^ led to a strong decrease in L5s (0.61 ± 0.02 L5s per column; Figure 3V-W; P****<0.0001, one-way ANOVA with Dunn’s multiple comparisons test). Under these conditions we also observed Dcp1 positive apoptotic cells in the most proximal row of the lamina, which were never observed in controls (Figure 3—figure supplement 2) but were observed when L5 differentiation was blocked above (Figure 2B). Thus, xg^O^-derived Spi and Col4a1 are both necessary and sufficient to induce L5 differentiation. Altogether, we found that xg^O^ secrete multiple factors that lead to activation of the MAPK cascade in the proximal lamina to induce differentiation of L5s.”

2. Statistics: Could the authors please check their statistics? Non-parametric tests are used throughout the manuscript – are their data not normally distributed? We also recommend that the authors use graphs that show the distribution of their data. In the detailed reviews below are more specific comments regarding the statistics. Could the authors please pay attention to each of these points as they revise their manuscript?

Thank you for flagging that we had not reported our statistical analyses appropriately. We apologise for this and have made sure to explicitly state the statistical test performed for multiple and pairwise comparisons with the P-values as detailed by Reviewer 3. These are highlighted throughout the text with track-changes. As well, we have changed all our graphs to box and whisker plots showing the entire distribution of the data as well as the interquartile range, as recommended.

Much of the data in our manuscript are proportions generated from cell counts and, by definition, are limited to numerical values between 0 and 1 (inclusive). As such, as with count data (*i.e.* discrete numbers such as from cell counts), parametric statistics are generally inappropriate for proportion data because the data violate assumptions about normality (Douma and Weedon, 2019). Therefore, we used non-parametric tests throughout the manuscript except for Figure 1—figure supplement 1R where appropriate assumptions were met.

3. Enhancer trap lines: We were concerned about the validity of the enhancer trap lines. If any of them have been demonstrated in earlier studies to be faithful, could the authors please state that explicitly? For any lines that haven't been verified, can the authors please discuss the caveats also explicitly?

We apologise for not clarifying whether the enhancer traps used in our study had been validated previously. We have now added references and make explicit when enhancer traps have or have not been validated previously. Our study made use of enhancer trap lines for *bnl, ths, Col4a1, spi* and *Ddr,* of which enhancer traps used for *bnl, ths* and *Col4a1* had been validated previously: bnl-Gal4 [NP2211] (Chen and Krasnow, 2014; Kamimura et al., 2006; Spéder and Brand, 2014; Tamamouna et al., 2021), ths-Gal4 (Anllo and DiNardo, 2022; Wu et al., 2017) and Cg-Gal4 by (Hennig et al., 2006). Moreover, ths-Gal4 expression matched a previous report of *ths* expression reported by in situ hybridization (Franzdóttir et al., 2009).

The *spi* and *Ddr* enhancer traps we used were not previously validated. Therefore, to further substantiate these expression patterns we performed *in situ* hybridisation chain reaction (HCR) to evaluate *spi, Col4a1* and *Ddr* mRNA expression directly and confirmed that xg^O^ express *spi* and *Col4a1*, while *Ddr* was expressed in all cells of the lamina. We show these data in Figure 3—figure supplement 1H,I,Q in the revised manuscript.

HCR is a new form of fluorescent *in situ* hybridisation (FISH) used to detect nucleotides with enzyme-free fluorescent signal amplification (Choi et al., 2010; Dirks and Pierce, 2004). HCR employs initiators and amplifiers which when combined triggers a chain reaction of hybridisation events generating bright fluorescent amplified polymers (Choi et al., 2010; Dirks and Pierce, 2004). mRNA is detected by a probe set containing 20-40 short DNA probes. The initiator is split between pair of probes such that only those probes which hybridise to the target will generate a full initiator. The addition of metastable fluorescent hairpins (amplifiers) will bind to probes that colocalise the full initiator leading to the formation of bright puncta (Choi et al., 2018).

4. Spi mRNA in situ: The Spi mRNA in situ is ambiguous and removing it will not change the storyline. We recommend removing this data. Alternatively, the authors would need to address the concerns raised by the reviewers.

We have understood that the main concern around the *spi* HCR included in our manuscript relates to the fact that the signal detected in the nucleus was more abundant than just two puncta as would be expected from two sites of transcription.

The reviewers are correct that only two puncta corresponding to active sites of transcription would be expected in the nucleus when detected by single molecule FISH (smFISH). However, here we are not using smFISH but HCR with maximal amplification. This results in signal proportional to the relative abundance of transcripts (Choi et al., 2018; Trivedi et al., 2018) and as such all transcripts, including those moving away from the transcription site in the nucleus, are also detected by this method. Other groups who have used this method also report the same (Andrews et al., 2020; Duckhorn et al., 2022; Schwarzkopf et al., 2020; Zhuang et al., 2020). We used this form of HCR over single molecule HCR (smHCR or digital-HCR), which uses limited amplification (Trivedi et al., 2018), as these other methods require diffractionlimited spot detection, which would be very challenging in our system. We apologise for not explaining the HCR protocol sufficiently and have included more details in the Materials and methods.

In addition to using HCR to detect *spi* expression in xg^O^ in controls and when EGFR signalling is blocked in xg^O^, we now also provide new data to show *Col4a1* and *Ddr* expression using HCR, to lend support to enhancer traps that were not validated previously. We found that both *spi* and *Col4a1* expression in xg^O^ decreased when EGFR signalling was blocked in xg^O^ and provide single channel images in Figure 3 —figure supplement 1.

With this clarification, we hope the reviewers will reconsider the inclusion of these data as we feel it is important to show that xg^O^ express these ligands in an EGFR signalling-dependent manner, especially in light of the *spi* and *Col4a1* loss-of-function data detailed above. Nonetheless, if the reviewers still feel that these data should be removed from the manuscript, we will be happy to do so.

5. Overlap with Fernandes et al., 2017: The current manuscript rests heavily on the previous one. Many of the manipulations used in some of these assays were described earlier. The manner in which the manuscript is currently written may be laying equal emphasis on the previous discoveries and the novelty in this current one. For example, the screen to discover which EGFR ligand might be involved is described in two action-packed sentences : We suggest that the authors unpack and emphasise the novel aspects of this story ( xgo > L5 neuron specification > inhibition of the second set of L5s) to avoid giving the (false) impression that it is an overlap of their earlier one.

Thank you for appreciating the novelty of our story, despite it not coming across clearly in the previous version of our manuscript. We have taken these comments on board and restructured our manuscript as suggested to highlight its novelty, namely that outer chiasm giant glia induce L5 neuronal differentiation and that the tissue architecture and feedback from newly differentiating L5s ensures that the correct number of L5s are induced to differentiate. We have also removed supplementary text and included it in the main text as suggested by Reviewer 3, which has greatly improved the paper.

6. The 'un-dead' L5s: There is no guarantee that the ectopic lamina precursor cells found in Dronc mutant are identical to the cells that fail to differentiate to L5 due to insufficient MAPK signalling. The forced MAPK activation in Figure 1H, I does not clarify this point. Could the authors please revisit the discussion around these precursors to highlight this point?

We have highlighted this point in the discussion as follows:

Pages 12-13, lines 370-378:

“These ‘extra’ LPCs did not differentiate when apoptosis was blocked (Figure 5A,B) but generated ectopic L5s when forced to differentiate (Figure 4D and Figure 4—figure supplement 1D). Although we cannot rule out that preventing death using *Dronc* mutants may mis-specify the ‘extra’ cells and prevent them from differentiating, it is more likely that these ‘extra’ cells are specified as L5s, but that other mechanisms restricted their differentiation in *Dronc* mutants, as other lamina neuron types differentiated normally (Figure 5B). Thus, twice as many LPCs appear to be specified as L5s than undergo differentiation normally, which implies that a selection process to ensure the correct number of L5s develop is in place.”

7. Quantification of nuclear MAPKinase: It is not clear where dpMAPKinase is in the L5 neurons in Figure S3P, Q (nuclear or not). (The quantification in S3L is for xgo glia.) So, in the absence of quantification, it isn't clear how the authors can tell in which cells dpMAPkinase signal is. We recommend quantifying dMAPK in L5 in the two scenarios where Spi and Col4a are over-expressed in the xgo glia.

We now provide quantifications of the nuclear to cytoplasmic ratio of dpMAPK mean fluorescence intensity (MFI) in the most proximal row of LPCs when we block EGFR signalling in xg^O^ and then when we rescue with *spi* and *Col4a1* in the xg^O^. We used Dac expression to identify the most proximal row of lamina precursors and to segment the nucleus and then measured dpMAPK intensity within this area and in a set width around this region of interest to generate a nuclear:cytoplasmic ratio of dpMAPK. We found a significant increase in the nuclear to cytoplasmic ratio of dpMAPK MFI for the rescue experiments when compared to controls, suggesting that Spi and Col4a1 secreted by the xg^O^ activates MAPK signalling in the proximal LPCs.

Page 8, lines 240-244 (Figure3—figure supplement R-T):

“Moreover, expressing *spi* or *Col4a1* in xg^O^ in which EGFR signalling was blocked rescued dpMAPK signal in L5s, indicating that, when expressed in xg^O^, these ligands were sufficient to activate MAPK signalling in the proximal lamina (Figure3—figure supplement R-T; P**<0.005, P****<0.0001; one-way ANOVA with Dunn’s multiple comparisons Test).”

To clarify, in previous Figure S3L (now Figure 3—figure supplement 1K) we quantified the nuclear to cytoplasmic ratio of dpMAPK mean fluorescence intensity (MFI) in xg^O^ in *xg^O^>EGFR^DN^+s.spi* to rule out the trivial possibility that the rescue in L5 numbers was due to an autocrine response of the xg^O^ to Spi rather than due to Spi acting on LPCs.

References:

Akagawa H, Hara Y, Togane Y, Iwabuchi K, Hiraoka T. 2015. The role of the effector caspases drICE and dcp-1 for cell death and corpse clearance in the developing optic lobe in *Drosophila*. *Dev Biol* 1–15. doi:10.1016/j.ydbio.2015.05.013

Andrews TGR, Gattoni G, Busby L, Schwimmer MA, Benito-Gutiérrez È. 2020. Hybridization Chain Reaction for Quantitative and Multiplex Imaging of Gene Expression in Amphioxus Embryos and Adult Tissues BT – In Situ Hybridization Protocols In: Nielsen BS, Jones J, editors. New York, NY: Springer US. pp. 179–194. doi:10.1007/978-1-0716-0623-0_11

Anllo L, DiNardo S. 2022. Visceral mesoderm signaling regulates assembly position and function of the *Drosophila* testis niche. *Dev Cell* 57:1009-1023.e5. doi:https://doi.org/10.1016/j.devcel.2022.03.009

Chen F, Krasnow MA. 2014. Progenitor Outgrowth from the Niche in *Drosophila* Trachea Is Guided by FGF from Decaying Branches. *Science (80- )* 343:186– 189. doi:10.1126/science.1241442

Chen J, Xu N, Huang H, Cai T, Xi R. 2016. A feedback amplification loop between stem cells and their progeny promotes tissue regeneration and tumorigenesis. *eLife* 5:e14330. doi:10.7554/*eLife*.14330

Choi HMT, Chang JY, Trinh LA, Padilla JE, Fraser SE, Pierce NA. 2010. Programmable in situ amplification for multiplexed imaging of mRNA expression. *Nat Biotechnol* 28:1208–1212. doi:10.1038/nbt.1692

Choi HMT, Schwarzkopf M, Fornace ME, Acharya A, Artavanis G, Stegmaier J, Cunha A, Pierce NA. 2018. Third-generation in situ hybridization chain reaction : multiplexed, quantitative, sensitive, versatile, robust. *Development* 145:dev165753. doi:10.1242/dev.165753

Csordás G, Grawe F, Uhlirova M. 2020. Eater cooperates with Multiplexin to drive the formation of hematopoietic compartments. *eLife* 9:e57297. doi:10.7554/*eLife*.57297

Dirks RM, Pierce NA. 2004. Triggered amplification by hybridization chain reaction. *Proc Natl Acad Sci* 101:15275–15278. doi:10.1073/pnas.0407024101

Douma JC, Weedon JT. 2019. Analysing continuous proportions in ecology and evolution: A practical introduction to β and Dirichlet regression. *Methods Ecol Evol* 10:1412–1430. doi:https://doi.org/10.1111/2041-210X.13234

Duckhorn JC, Junker IP, Ding Y, Shirangi TR. 2022. Combined in Situ Hybridization Chain Reaction and Immunostaining to Visualize Gene Expression in WholeMount *Drosophila* Central Nervous Systems JuliaBehavioral Neurogenetics. Springer Nature. doi:10.1007/978-1-0716-2321-3

Franzdóttir SR, Engelen D, Yuva-Aydemir Y, Schmidt I, Aho A, Klämbt C. 2009. Switch in FGF signalling initiates glial differentiation in the *Drosophila* eye. *Nature* 460:758–761. doi:10.1038/nature08167

Hennig KM, Colombani J, Neufeld TP. 2006. TOR coordinates bulk and targeted endocytosis in the *Drosophila melanogaster* fat body to regulate cell growth. *J Cell Biol* 173:963–974. doi:10.1083/jcb.200511140

Huang Z, Kunes S. 1998. Signals transmitted along retinal axons in *Drosophila*: Hedgehog signal reception and the cell circuitry of lamina cartridge assembly. *Development* 125:3753–64. doi:10.1016/s0092-8674(00)80094-x

Huang Z, Kunes S. 1996. Hedgehog, transmitted along retinal axons, triggers neurogenesis in the developing visual centers of the *Drosophila* brain. *Cell* 86:411–22. doi:10.1016/S0092-8674(00)80114-2

Kamimura K, Koyama T, Habuchi H, Ueda R, Masu M, Kimata K, Nakato H. 2006. Specific and flexible roles of heparan sulfate modifications in *Drosophila* FGF signaling. *J Cell Biol* 174:773–778. doi:10.1083/jcb.200603129

Louradour I, Sharma A, Morin-Poulard I, Letourneau M, Vincent A, Crozatier M, Vanzo N. 2017. Reactive oxygen species-dependent Toll/NF-κB activation in the *Drosophila* hematopoietic niche confers resistance to wasp parasitism. *eLife* 6:e25496. doi:10.7554/*eLife*.25496

Morante J, Vallejo DM, Desplan C, Dominguez M. 2013. Conserved miR-8/miR-200 Defines a Glial Niche that Controls Neuroepithelial Expansion and Neuroblast Transition. *Dev Cell* 27:174–187. doi:10.1016/j.devcel.2013.09.018

Pastor-Pareja JC, Xu T. 2011. Shaping Cells and Organs in *Drosophila* by Opposing Roles of Fat Body-Secreted Collagen IV and Perlecan. *Dev Cell* 21:245–256. doi:10.1016/j.devcel.2011.06.026

Schwarzkopf M, Choi HMT, Pierce NA. 2020. Multiplexed Quantitative In Situ Hybridization for Mammalian Cells on a Slide: qHCR and dHCR Imaging (v3.0) BT – In Situ Hybridization Protocols In: Nielsen BS, Jones J, editors. New York, NY: Springer US. pp. 143–156. doi:10.1007/978-1-0716-0623-0_9

Spéder P, Brand AH. 2014. Gap junction proteins in the blood-brain barrier control nutrient-dependent reactivation of *Drosophila* neural stem cells. *Dev Cell* 30:309–321. doi:10.1016/j.devcel.2014.05.021

Sugie A, Umetsu D, Yasugi T, Fischbach KF, Tabata T. 2010. Recognition of pre- and postsynaptic neurons via nephrin/NEPH1 homologs is a basis for the formation of the *Drosophila* retinotopic map. *Development* 137:3303–3313. doi:10.1242/dev.047332

Tamamouna V, Rahman MM, Petersson M, Charalambous I, Kux K, Mainor H, Bolender V, Isbilir B, Edgar BA, Pitsouli C. 2021. Remodelling of oxygentransporting tracheoles drives intestinal regeneration and tumorigenesis in *Drosophila*. *Nat Cell Biol* 23:497–510. doi:10.1038/s41556-021-00674-1

Trivedi V, Choi HMT, Fraser SE, Pierce NA. 2018. Multidimensional quantitative analysis of mRNA expression within intact vertebrate embryos. *Development* 145:dev156869. doi:10.1242/dev.156869

Tsruya R, Schlesinger A, Reich A, Gabay L, Sapir A, Shilo B. 2002. Intracellular trafficking by Star regulates cleavage of the *Drosophila* EGF receptor ligand Spitz. *Genes Dev* 12:222–234. doi:10.1101/gad.214202.fate

Umetsu D, Murakami S, Sato M, Tabata T. 2006. The highly ordered assembly of retinal axons and their synaptic partners is regulated by Hedgehog/Singleminded in the *Drosophila* visual system. *Development* 133:791–800. doi:10.1242/dev.02253

Urban S, Lee JR, Freeman M. 2002. A family of rhomboid intramembrane proteases activates all *Drosophila* membrane-tethered EGF ligands. *EMBO J* 21:4277– 4286. doi:10.1093/emboj/cdf434

Wu B, Li J, Chou Y, Luginbuhl D, Luo L. 2017. Fibroblast growth factor signaling instructs ensheathing glia wrapping of *Drosophila* olfactory glomeruli. *Proc Natl Acad Sci* 114:7505–7512. doi:10.1073/pnas.1706533114

Yogev S, Schejter ED, Shilo B-Z. 2008. *Drosophila* EGFR signalling is modulated by differential compartmentalization of Rhomboid intramembrane proteases. *EMBO J* 27:1219–1230. doi:https://doi.org/10.1038/emboj.2008.58

Zhuang P, Zhang H, Welchko RM, Thompson RC, Xu S, Turner DL. 2020. Combined microRNA and mRNA detection in mammalian retinas by in 6situ hybridization chain reaction. *Sci Rep* 10:351. doi:10.1038/s41598-019-57194-0